# Refining models of archaic admixture in Eurasia with *ArchaicSeeker 2.0*

Kai Yuan[1,9], Xumin Ni[2,9], Chang Liu[1,9], Yuwen Pan[1,9], Lian Deng[3,9], Rui Zhang[1,9], Yang Gao[1,4], Xueling Ge[1], Jiaojiao Liu[3], Xixian Ma[1], Haiyi Lou[3], Taoyang Wu [5] & Shuhua Xu [1,3,4,6,7,8✉]

We developed a method, *ArchaicSeeker 2.0*, to identify introgressed hominin sequences and model multiple-wave admixture. The new method enabled us to discern two waves of introgression from both Denisovan-like and Neanderthal-like hominins in present-day Eurasian populations and an ancient Siberian individual. We estimated that an early Denisovan-like introgression occurred in Eurasia around 118.8–94.0 thousand years ago (kya). In contrast, we detected only one single episode of Denisovan-like admixture in indigenous peoples eastern to the Wallace-Line. Modeling ancient admixtures suggested an early dispersal of modern humans throughout Asia before the Toba volcanic super-eruption 74 kya, predating the initial peopling of Asia as proposed by the traditional Out-of-Africa model. Survived archaic sequences are involved in various phenotypes including immune and body mass (e.g., *ZNF169*), cardiovascular and lung function (e.g., *HHAT*), UV response and carbohydrate metabolism (e.g., *HYAL1/HYAL2/HYAL3*), while "archaic deserts" are enriched with genes associated with skin development and keratinization.

[1] Key Laboratory of Computational Biology, Shanghai Institute of Nutrition and Health, University of Chinese Academy of Sciences, Chinese Academy of Sciences, Shanghai 200031, China. [2] Department of Mathematics, School of Science, Beijing Jiaotong University, Beijing 100044, China. [3] State Key Laboratory of Genetic Engineering, Collaborative Innovation Center of Genetics and Development, Center for Evolutionary Biology, School of Life Sciences, Fudan University, Shanghai 200433, China. [4] School of Life Science and Technology, ShanghaiTech University, Shanghai 201210, China. [5] School of Computing Sciences, University of East Anglia, Norwich NR4 7TJ, UK. [6] Center for Excellence in Animal Evolution and Genetics, Chinese Academy of Sciences, Kunming 650223, China. [7] Henan Institute of Medical and Pharmaceutical Sciences, Zhengzhou University, Zhengzhou 450052, China. [8] Human Phenome Institute, Fudan University, Shanghai 201203, China. [9]These authors contributed equally: Kai Yuan, Xumin Ni, Chang Liu, Yuwen Pan, Lian Deng, Rui Zhang. ✉email: xushua@fudan.edu.cn

The publication of a draft sequence of the Neanderthal genome in 2010 was followed by a decade of efforts to elucidate the complex admixture landscape between archaic and modern humans. Various archeological and genetic studies have conclusively demonstrated that the anatomically modern human (AMH) coexisted with several distinct archaic hominins, including Neanderthals[1–3], Denisovans[4], and perhaps others[5]. Due to a combination of factors that may include climate changes[6,7], diseases[8,9], and activities of the modern humans[10], the archaic hominins ceased to exist before 30 thousand years ago (kya). Nevertheless, analysis of the DNA from these extinct archaic hominins provides strong evidence of gene flow from archaic hominins to modern humans[1,4]. Indeed, several studies documented the extent of variation in archaic ancestry among human populations and estimated that most of the non-African genomes carry approximately 1–3% archaic hominin sequences[2–4,11,12]. However, it remains inconclusive as to when, where, and how this small amount of archaic DNA was introduced into the present-day human genomes[13–20].

Several methods have been developed for identifying archaic sequences in present-day human genomes[1,2,13,16–18,21–25]. One notable example is Patterson's $D$-statistic[21], also commonly known as the ABBA-BABA test[22], which played an important role in the initial confirmation of the inbreeding between modern humans and archaic hominins[1]. The main idea of this method is to compare allele-sharing between two populations and an outgroup, which also inspired the development of several methods, including the original version of *ArchaicSeeker*[23]. However, one common limitation of them is the difficulty to determine the precise boundaries of introgression segments. To address this shortcoming, several methods are proposed under a unifying postulate that genomic regions with introgressed sequences should have distinct patterns compared with non-introgressed regions. Based on the type of information used to delineate introgression segments, these methods could be roughly classified into the following two groups. The first group uses populational-level information to detect introgressed sequences[11,13,16,19]. For instance, $S^*$ series methods[11,16,19,26] and *Sprime*[13] were developed based on the observation that linkage disequilibrium (LD) in the introgressed regions is higher than that in non-introgressed genomic regions. These methods typically have higher power and accuracy than previous methods in exploring the inbreeding between modern humans and archaic hominins. The other group of methods analyzes the introgression on the individual or haplotype level[2,17,18,25,27]. These methods are generally based on sequential pattern recognition models such as the Hidden Markov Model (HMM)[2,18,23,25,27] and Conditional Random Field[17]. While most of these methods detect introgressed sequences by comparing the test introgressed genome with archaic and African genomes, Hu et al.[25] and Skov et al.[18] used the information of non-AMH marker density to identify the introgressed sequences.

Despite the aforementioned tremendous efforts, modeling archaic–hominin admixture remains challenging. Compared with detecting introgressed sequences, admixture history inference is intrinsically more difficult. Consequently, fewer methods have been proposed and developed. Furthermore, the majority of currently available methods suffer from one or more of the following three limitations. The first limitation is being too computationally expensive, often resulting from the use of large-scale simulations to fit the introgression history[17,19,26]. The second limitation is that some methods were designed specifically for some scenarios or targeted at a particular population thus are not necessarily suitable for general populations[14,15,17,26]. The third limitation is the loss of information on shorter segments. Two such examples are *Sprime*[13] and Jacobs' method[14], albeit providing a more general approach than many others. These two methods infer the number of introgression waves by classifying introgressed sequences into different groups according to the number of nucleotide differences between candidate introgressed sequences and an archaic reference genome. However, classification based on nucleotide differences is often not sufficient to assign introgressed sequences into different admixture events in history. For instance, under the scenarios in which a hominin group was admixed with more than one distinct ancestral archaic hominin or the reference (sequenced) genome per se was admixed, these methods might overestimate the number of introgression waves. In addition, these two methods require extremely stringent filtration procedures to remove shorter segments of archaic ancestry, and hence in some applications, only a few sequences or chunks longer than hundreds of kilobases were used for the final history inference. In principle, admixture history could be better inferred from admixed tract lengths than from nucleotide differences. Indeed, Skov's method[18] used the average archaic tract length to estimate introgression time, but this method only allowed modeling single-wave introgression without distinguishing the introgression from different archaic lineages. Some other methods such as *Legofit*[28] and *Admixtools*[29] were proposed to reconstruct the multiple-wave introgression history using nucleotide information. For instance, *Legofit* was used to show that ancestors of Neanderthals and Denisovans may have interbred with Eurasian predecessors[30].

In this work, we propose a generalized method called *ArchaicSeeker 2.0* to simultaneously detect ancient sequences in the modern human genomes that are derived from archaic hominins and infer the introgression history. This method implements an HMM to describe the genome of mixed ancestries derived from archaic and modern humans, a likelihood-based approach to trace ancestral sources of introgressed sequences, and a general discrete admixture model as implemented in the *MultiWaver* suite of methods[31–33] to infer multiple-wave complex admixture history (Fig. 1). *ArchaicSeeker 2.0* avoids massive simulations and artificial filtration of the surviving archaic sequences inferred from present-day human genomes. Furthermore, *ArchaicSeeker 2.0* enables us to detect introgressed sequences from any modern human genome, determine ancestral source groups including those archaic hominins which are currently unknown, infer the ancestry of each introgressed sequence, and reconstruct the population admixture history. This is demonstrated here by applications of *ArchaicSeeker 2.0* to detect archaic sequences in worldwide populations (Supplementary Note 1) and to reconstruct archaic–hominin admixture history in Eurasia and Oceania. Our results also suggest that there was more than one wave of dispersal of modern humans from Africa to Eurasia.

## Results

**Power and precision of *ArchaicSeeker 2.0*.** *ArchaicSeeker 2.0* is a method to simultaneously detect archaic introgression sequences and reconstruct archaic–modern human admixture history (see "Methods" and Supplementary Note 2). We conducted simulation studies under a variety of admixture scenarios to evaluate its performance, especially its power and accuracy (Supplementary Note 3). The results showed that the precision and the true-positive rate (TPR) were both high and the false-positive rate (FPR) was low. We compared the *ArchaicSeeker 2.0* results with the ground-truth introgressed sequences in simulation data, using three different schemes: a length-based direct comparison, single-nucleotide polymorphism (SNP)-based direct comparison, and segment-based comparison.

The lengths of the introgressed sequences identified by *ArchaicSeeker 2.0* were directly compared with those of the simulated ground-truth results and measured on the basis of bp

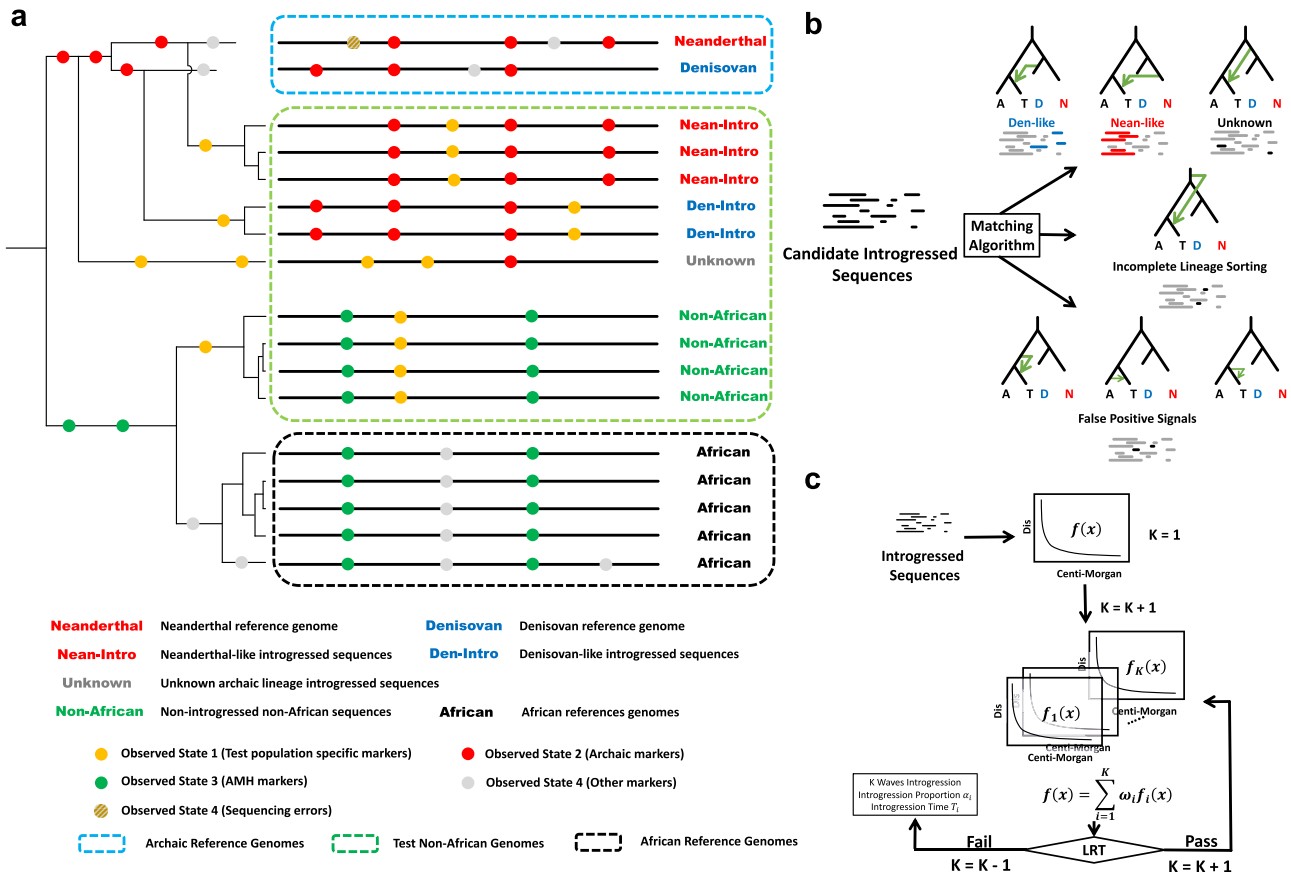

**Fig. 1 *ArchaicSeeker 2.0* schematic. a** Seeking algorithm. The light blue dashed box includes archaic reference genomes, the light green dashed box includes tested non-African human genomes, and the black dashed box includes African reference genomes. Dots in different colors stand for mutations of four different observation states in the HMM. **b** Matching algorithm. After we got the candidate introgressed sequences, we matched them to the seven topologies and found the best-matched one. Each topology corresponds to introgression for one specific lineage, incomplete lineage sorting, or a false-positive signal. **c** Reconstructing introgression history. For those introgressed sequences from one specific lineage, the length distribution was used to reconstruct the introgression history. A likelihood ratio test was used to find the most likely number of introgression events (*K*) and the software estimated introgression time and the proportion of each introgression event. AMH, anatomically modern human.

concordance. The median value of the precision was 93.0% (95% CI, 89.4–95.9%), that of the TPR was 90.4% (95% CI, 84.1–94.1%), and that of the FPR was 0.14% (95% CI, 0.07–0.22%) (Supplementary Note 4.1.1).

We also evaluated our methods by comparing the number of different SNPs. The results of the SNP-based direct comparison are overall similar to those of the length-based comparison (Supplementary Note 4.1.2). However, the evaluation results were further improved when only the non-AMH ancestry informative markers (AIMs, monomorphic in Africans) were considered. The median value of the precision was 99.3% (95% CI, 98.9–99.6%), that of the TPR was 93.7% (95% CI, 87.1–96.5%), and that of the FPR was 0.14% (95% CI, 0.07–0.24%) (Supplementary Note 4.1.2). This shows that *ArchaicSeeker 2.0* performs rather well with the non-AMH AIMs.

In the scenarios of deep divergent or unknown archaic lineage introgression, the precision and FPR of *ArchaicSeeker 2.0* results were good and stable. However, the TPR decreased slightly when the divergent time between known archaic lineages and the truly introgressed unknown lineage, $T_{\text{split}}$, increased. When $T_{\text{split}}$ was set to 610 kya, the TPR dropped to 81.9% (95% CI, 80.0–83.5%). Although the performance was not as good as that in recent divergent scenarios, the TPR was still greater than 80% and the precision was about 93% (Supplementary Note 4.1.1). These results demonstrate the power and robustness in detecting unknown introgression.

The precision in the identification of the introgressed segments is crucial to the admixture history inference. We further evaluated the performance of *ArchaicSeeker 2.0* in this respect. The precision of introgressed segments was estimated as the ratio of the number of correctly inferred segments to the total number of segments obtained from *ArchaicSeeker 2.0* analysis, where a segment was deemed correctly identified if its overlap with the ground-truth segments was more than 80% when measured by segment length. As expected, segments of smaller size tended to be subjected to many more stochastic errors (Supplementary Note 4.1.3). When the segments shorter than 15 kb were excluded, almost 90% of the remaining segments were correctly detected under most of the scenarios (Supplementary Note 4.1.3). To obtain an accurate distribution of the segment length, we removed those segments shorter than 15 kb in the following history analysis, but these effects would be taken into account in our history inference methods[31]. In all scenarios, the distribution of inferred segments and that of the ground-truth segments were nearly identical (Supplementary Note 4.2.1). The accurate estimation of segment length distribution makes the following introgression history inference possible. The matching algorithm enabled us to determine the ancestry of each inferred segment. More than 80% of the segments were matched to the proper ancestral lineages. The accuracy might be different for different scenarios; more recent divergence from the known archaic

hominins showed higher matching accuracy (Supplementary Note 4.1.4).

To evaluate the performance of *ArchaicSeeker 2.0* in inferring admixture history, we conducted two studies, one based on simulated ground-truth segments and one based on de novo detected segments by *ArchaicSeeker 2.0*. With simulated ground-truth segments, *ArchaicSeeker 2.0* succeeded in modeling the introgression history in all one-wave admixture scenarios and all two-wave admixture with introgression from two different archaic lineages scenarios (Supplementary Note 4.2.2). In the case of a two-wave admixture with introgression from one single archaic lineage, *ArchaicSeeker 2.0* succeeded in modeling 41 of 48 admixture scenarios. Under scenarios of more complicated admixture, for example, where the ancestry contribution of the two introgression waves was very different and the introgression time of two waves was closer, *ArchaicSeeker 2.0* tended to simplify the model and suggested a single-wave admixture. In summary, *ArchaicSeeker 2.0* has good performance in modeling archaic admixture history under various scenarios with both the ground-truth segments and the *de novo* detected segments. In particular, it accurately estimated the number of introgression events and the corresponding ancestry contribution as well as the admixture time (Supplementary Note 4.2.2 and 4.2.3).

We further evaluated the performance of *ArchaicSeeker 2.0* with the *de novo* detected segments to assess the influence of the errors in the identification of introgressed segments, which would be inevitable in real data analysis (Supplementary Note 4.2.3). In 144 different simulation scenarios, *ArchaicSeeker 2.0* correctly inferred 122 admixture models, i.e., a success rate of 84.7%. In scenarios of introgression from unknown archaic groups and different introgression times, *ArchaicSeeker 2.0* perfectly inferred all the 24 admixture models, i.e., a success rate of 100%. Under 18 scenarios where archaic lineages had different divergence times, our method failed only once by suggesting an extra introgression event (a success rate of 94.4%). Under some more complex scenarios of different ancestry contributions of different introgression events, *ArchaicSeeker 2.0* had a lower performance, which was likely caused by the introgressed segment detection error in the cases that have an admixture event of extremely small ancestry contributions. When modeling two introgression events contributed by one archaic lineage, *ArchaicSeeker 2.0* had a success rate of 91.7% (44 out of 48). The four failed inferences were in those scenarios with two relatively close introgression events where one played a dominant role. Under scenarios of introgression from two archaic hominin groups, *ArchaicSeeker 2.0* showed a lower success rate of 63.9% (23 of 36). In most of the failed cases, our method suggested an extra introgression event. However, the extra introgression event could be possibly screened in all of the 13 failed cases by bootstrapping analysis, because we observed the introgression proportions of the extra event were usually small and unstable. In addition, the extra introgression event did not affect the modeling of the "true" introgressed event.

In summary, *ArchaicSeeker 2.0* showed high precision (~93%), high TPR (~90.4%), and low FPR (~0.14%) in detecting archaic sequences under different introgression scenarios (see Supplementary Notes 3 and 4 for details). It has satisfying performance in reconstructing introgression models in various simulated scenarios (Supplementary Notes 4.2.2 and 4.2.3). We therefore explicitly demonstrated the power and robustness of our method in identifying archaic introgressed sequences and modeling archaic human admixture history.

**Genomic distribution of surviving archaic sequences**. We applied *ArchaicSeeker 2.0* to analyze whole-genome sequencing data of global diverse population samples included in the 1000

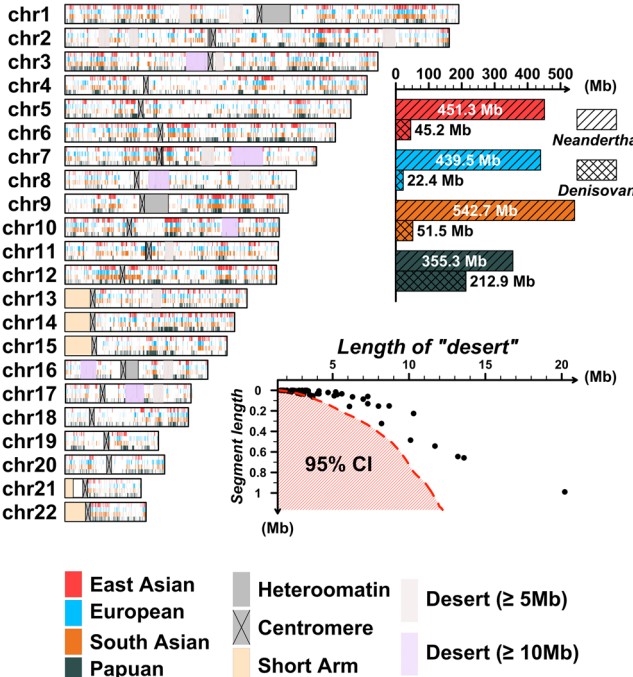

**Fig. 2 The landscape of archaic introgression in non-Africans.** This figure demonstrates the genomic coverage of archaic introgression in four different continental/regional populations. To eliminate the influence of sample size, we set a minimal introgression frequency threshold of each position as 0.02. Regions with introgression frequency greater than 0.02 were defined as introgression-covered regions. We plotted the introgression-covered regions on the genome, where four different colors stand for the four different continental/regional populations, respectively. Red stands for East Asian populations; light blue stands for European populations; orange stands for South Asian populations; and dark blue stands for the Papuan population. The boxplot shows the total introgression-covered length of Neanderthal and Denisovan ancestry for the four different populations. We also performed a test of introgression "deserts." We first divided the genome into thousands of 100-kb bins. Then, we obtained the empirical distribution of introgression-covered length. A two-tailed test was performed to find those genomic regions with extremely rare introgression segments. For the detailed statistical method, please refer to Supplementary Note 5. The red shadow indicates the 95% confidence interval and each black dot stands for one introgression "desert." We also plotted the long "desert" on the genome with two different colors. Purple on the genome stands for "deserts" longer than 10 Mb, and brown stands for "deserts" longer than 5 Mb.

Genomes Project (KGP)[34], the Simons Genome Diversity Project (SGDP)[35], and the Estonian Biocentre Human Genome Diversity Panel (EGDP)[36] (Supplementary Notes 1 and 5).

We identified 451.3 Mb covered by Neanderthal-like introgression sequences in East Asian genomes, 439.5 Mb in European genomes, 542.7 Mb in South Asian genomes, and 355.3 Mb in Papuan genomes (Fig. 2). These coverages were corrected for sample size by counting the regions with more than 2% local introgression frequency. The Neanderthal-like sequence coverage is the highest among South Asian populations, while the coverage in Papuan genomes is the lowest. We also detected 45.2 Mb covered by Denisovan-like introgressed sequences in East Asian genomes, 22.4 Mb in European genomes, 51.5 Mb in South Asian genomes, and 212.9 Mb in Papuan genomes (Fig. 2). Obviously, Papuan genomes possessed the highest coverage of Denisovan-like introgressed sequences. East Asian and South Asian genomes had a higher Denisovan-like introgressed sequence coverage compared with European genomes. These observations indicated

the Denisovan-like introgression events that happened in Papua New Guinea were different from those that happened in Eurasia. We will further discuss the relationship between introgressed sequence coverage and introgression history.

The genomic location of introgression sequences was not evenly distributed in the genome (Fig. 2); 84 introgression "deserts" were found in autosomes, with six regions being longer than 10 Mb (Supplementary Note 5.4). *FOXP2*, which is essential to speech and language development[16,17,19,37], is located in the longest (20.2 Mb) desert, on Chromosome 7 (Fig. 2 and Supplementary Note 5.4).

**The landscape of archaic admixture in Eurasia**. The application of *ArchaicSeeker 2.0* to present-day human population data allowed us to discern two waves of gene flow from Denisovan-like hominins, and two waves of gene flow from Neanderthal-like hominins into non-African populations except for indigenous people living in islands east of the Wallace Line (Fig. 3). The first wave of introgression contributed 0.05–0.08% Denisovan-like ancestry and 0.31–0.67% Neanderthal-like ancestry, which is shared by all studied Eurasian populations. We estimated that this first wave of Denisovan-like introgression occurred around 118.8–94.0 kya and that the first wave of Neanderthal-like introgression occurred around 58.8–49.1 kya (Tables 1 and 2 and Supplementary Notes 5.2 and 5.3).

Notably, the second wave of Denisovan admixture occurred independently in East Asia and South Asia according to the distinct ancestry-sharing patterns observed in the two regions (Fig. 4a), although the estimation of introgression time was similar between East Asia and South Asia. In East Asia, the second wave of Denisovan-like introgression contributed 0.04–0.07% of the East Asian gene pool and took place 48.1–37.5 kya. In South Asia, the second wave of Denisovan-like introgression contributed 0.03–0.05% of the South Asian gene pool and took place 56.7–47.8 kya. In contrast, European populations received the second wave of Denisovan-like ancestry (~0.01%) indirectly from some Asian populations (Supplementary Notes 5.2 and 5.3), which will be further discussed in the following sections.

Our analysis also documented the extent of variation in the second wave of Neanderthal-like introgression among populations. We estimated that it contributed to 0.76–1.04% of the East Asian gene pool, to 0.48–0.76% of the South Asian gene pool, and to 0.66–0.80% of the European gene pool, which occurred around 37.5–33.0 kya (Table 2) when the ancestral populations of Asians and Europeans had already diverged from each other. The archaic ancestry-sharing analysis of populations from different continents showed more archaic ancestry-sharing between intra-continental populations and less archaic ancestry-sharing between inter-continental populations (Fig. 4a, b and Supplementary Note 5.3). These results indicate that there were both independent and shared introgression events among Eurasian populations.

**Analysis of a 45,000-year-old ancient Siberian confirmed the ancient introgression in present-day Eurasian populations**. We further applied *ArchaicSeeker 2.0* to the analysis of an ancient Siberian (Ust'-Ishim), who died 45,000 years ago[38]. The analysis of the Ust'-Ishim confirmed our results, i.e., a weak Denisovan-like introgression (~0.04%) was detected at around 147.6–92.3 kya (Table 3), similar to the first-wave introgression seen in the present-day Eurasian populations (118.8–94.0 kya). Interestingly, we also detected two waves of Neanderthal-like introgression in the Ust'-Ishim genome: a recent one (1.41–1.57%) that occurred 61.4–57.8 kya and a weaker and more ancient one (0.04–0.20%) that occurred 204.1–95.6 kya. The

recent Neanderthal-like introgression in the Ust'-Ishim was consistent with the first-wave introgression shared by present-day Eurasian populations (58.8–49.1 kya), although the percentage of Neanderthal-like ancestry was much higher than that in present-day populations. In present-day Eurasians, only 0.31–0.67% of Neanderthal-like sequences from the shared first-wave introgression remain, which might be due to recent negative selection against the introgressed sequences[38,39]. The drift and sampling error probably also contributed to a higher percentage of Neanderthal-like ancestry in Ust'-Ishim. There could also be some uncertainty with respect to the ancient wave of Neanderthal-like introgression identified in the Ust'-Ishim, which was likely due to the low power of the method in case of shorter segments, and which was also affected more by sequencing errors of the Ust'-Ishim genome and the extremely small sample size, i.e., a single individual genome of Ust'-Ishim.

**Differentiated introgression history between West and East of the Wallace Line**. Strikingly, despite a shared history of Neanderthal-like introgression in all studied Eurasian populations, Denisovan-like introgression in the Papuan population was different from that in other Eurasian populations. Intriguingly, there was an obvious elevation of Denisovan-like ancestry proportion (0.41–0.73%) (Table 1) in some indigenous populations living east of the Wallace Line and in the Batak from the Palawan island in the Philippines (Fig. 3), but the Neanderthal-like ancestry proportion was similar on both sides of the Wallace Line (West: 1.02–1.43%; East: 1.32–1.54%) (Table 1 and Fig. 3). The Batak are considered by anthropologists to be closely related to the Aeta of Central Luzon, east of the Wallace Line. In more detail, there were two waves of Denisovan-like introgressions in populations west of the Wallace Line; both occurred approximately at the same moment as that in Eurasia. However, there was only one single wave of Denisovan-like introgression in the Papuan population who lived east of the Wallace Line, which occurred 64.0–61.9 kya (Table 2). The Neanderthal-like introgression in Southeast Asian and Oceanian populations was similar to that seen in Eurasian populations, with respect to both time and ancestry contribution (Table 2). The diversity of Denisovan-like ancestry in Papuans is much higher than that in Eurasians, and the Denisovan-like ancestry-sharing within the Papuan population was lower than that within Eurasian populations (Fig. 4a, c and Supplementary Note 5.3). Together with the observation of the extremely high proportion of Denisovan-like sequences, these results suggest that there must have been an independent introgression from a Denisovan-like archaic group into populations living east of the Wallace Line.

On the contrary, the diversity of Neanderthal-like ancestry in the Papuan population was lower than that in Eurasian populations, and the Neanderthal-like ancestry-sharing within the Papuan population was much higher than that within Eurasian populations (Figs. 4b, d and Supplementary Note 5.3). These results suggested that the Neanderthal-like lineage was probably a result of an indirect introgression by gene flow to the Papuan population from surrounding modern human populations with Neanderthal-like ancestry.

The Denisovan-like introgression events in the Papuan population happened earlier (64.0–61.9 kya) than two waves of Neanderthal-like introgression (first wave: 61.7–53.0 kya; second wave: 35.2–28.9 kya). If there was only one single migration of modern humans across the Wallace Line, the Neanderthal-like introgression would be expected to take place after the modern humans arrived in Oceania, and the Neanderthal introgression in Oceania would be expected to be independent of that in Eurasia since Denisovan-like introgression took place independently in the Papuan population located east of the Wallace Line. However,

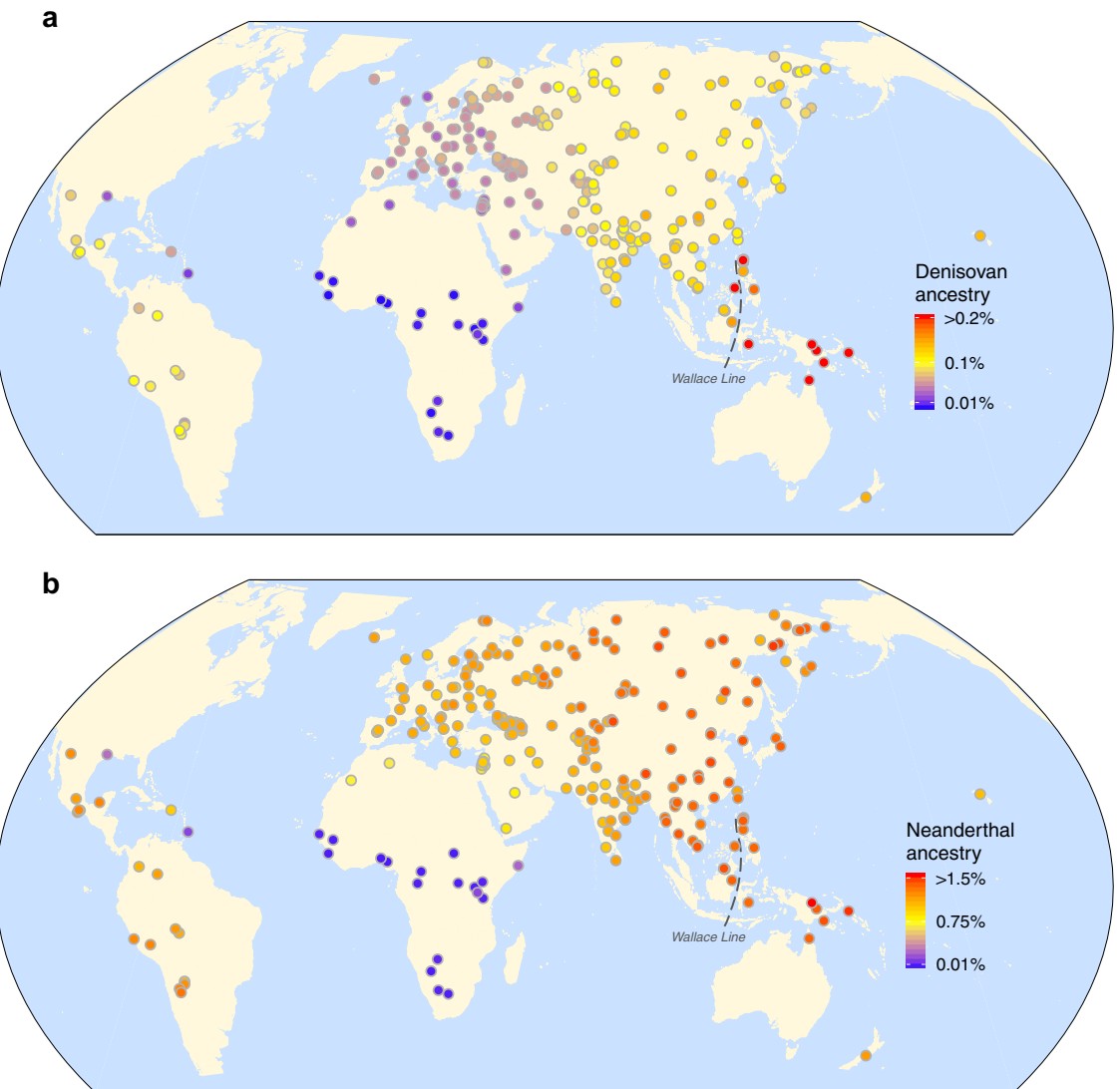

**Fig. 3 Archaic introgression in modern humans across the world. a** Average proportions of Denisovan introgression in modern humans. Proportions of >0.2% are presented as 0.2% for visualization. **b** Average proportions of Neanderthal introgression in modern humans. Proportions of >1.5% are presented as 1.5% for visualization.

**Table 1 Estimated introgression proportions (%) on different continents/regions.**

| Continent/region | Denisovan-like | Neanderthal-like |
|---|---|---|
| Africa | 0.00 (0.00–0.01)[a] | 0.01 (0.00–0.12) |
| America | 0.09 (0.01–0.10) | 1.21 (0.12–1.31) |
| Central Asia/Caucasus/Siberia | 0.10 (0.04–0.14) | 1.28 (0.97–1.48) |
| East Asia | 0.12 (0.08–0.14) | 1.37 (1.10–1.45) |
| Oceania/North Philippine Negrito (Aeta, Agta, and Batak) | 0.61 (0.41–0.73) | 1.39 (1.32–1.54) |
| SoutheastAsia | 0.13 (0.10–0.24) | 1.36 (1.02–1.43) |
| South Asia | 0.10 (0.06–0.13) | 1.11 (0.90–1.28) |
| West Eurasia | 0.05 (0.03–0.09) | 1.09 (0.73–1.33) |

[a]Introgression proportions were estimated with 1000 Genomes Project, Simons Genome Diversity Project, and Estonian Biocentre Human Genome Diversity Panel data. Numbers indicate the median value of the introgression proportion of that continent/region and numbers in parentheses are range from minimum to maximum (Supplementary Note 5.1).

the Neanderthal-like ancestry-sharing analysis invalidated these hypotheses. A reasonable interpretation of the archaic–modern human admixture patterns in Eurasia and Oceania would be the following. There was a second wave of "Out of Africa" migration of modern humans, who were admixed with Neanderthal-like sequences and migrated from Asia to Oceania (Figs. 5 and 6).

**Denisovan-like introgression in European populations.** Previous studies suggested that there is little evidence of Denisovan-like gene flow into European populations[14,35]. In our analysis, however, the ancestry-sharing ratio of Denisovan-like sequences across European populations was much higher than that between any non-European populations in Eurasia,

**Table 2 Inference and timing of multiple-wave introgression history.**

| Continent/region | Denisovan-like | | Neanderthal-like | |
| --- | --- | --- | --- | --- |
| | **Ancient wave** | **Recent wave** | **Ancient wave** | **Recent wave** |
| Europe[a] | 118.8-101.6[b] (0.05-0.06) | 46.8-22.7 (~0.01) | 58.8-54.7 (0.31-0.45) | 35.2-33.1 (0.66-0.80) |
| South Asia | 110.8-101.9 (0.07-0.08) | 56.7-47.8 (0.03-0.05) | 58.2-53.2 (0.45-0.67) | 37.5-33.1 (0.48-0.76) |
| East Asia | 113.3-94.0 (0.05-0.08) | 48.1-37.5 (0.04-0.07) | 56.9-49.1 (0.37-0.62) | 34.7-33.0 (0.76-1.04) |
| Oceania | 64.0-61.9 (0.73) | | 61.7-53.0 (0.61-1.07) | 35.2-28.9 (0.48-0.93) |

[a]Results of Iberian populations in Spain (IBS) did not support the two-wave Denisovan-like introgression model, so the Denisovan-like introgression model of Europeans did not support the results of IBS (Supplementary Note 5.2).
[b]Introgression history was estimated with European, South Asian, and East Asian data from the 1000 Genomes Project and Papuan data from the Simons Genome Diversity Project. The numbers are the minimum of the lower bound to the maximum of the upper bound introgression time of populations in the indicated continents/regions and are measured thousands of years ago (kya). Generation time was taken as 30 years (Supplementary Note 5.2). Numbers in the brackets are the range of minimum to maximum introgression proportion of populations in the indicated continent/region (Supplementary Note 5.2). Introgression proportion is displayed as a percentage (%).

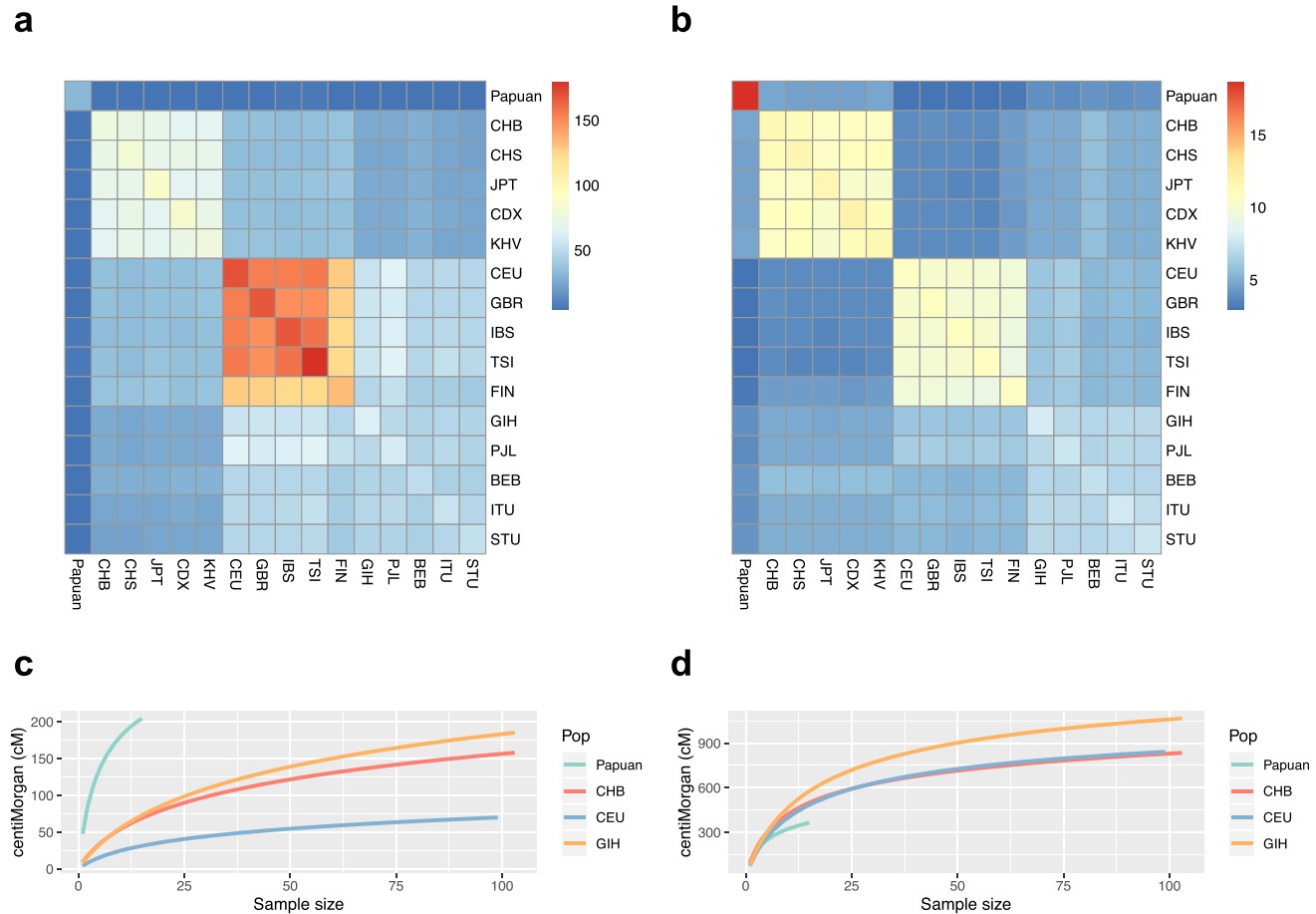

**Fig. 4 Ancestry-sharing ratio and introgression diversity. a** Ancestry-sharing ratio of Denisovan-like introgressed sequences. **b** Ancestry-sharing ratio of Neanderthal-like introgressed sequences. The heatmap of genomic introgression position shares statistics among worldwide populations. Warm colors indicate more sharing of the introgression position and cold colors indicate less sharing. **c** Introgression diversity of Denisovan-like introgressed sequences. **d** Introgression diversity of Neanderthal-like introgressed sequences. The y-axis shows the total introgressed sequence length and the x-axis shows the sample size. To avoid sampling errors, 10,000 permutations were performed. CHB Han Chinese from Beijing, China; CHS Han Chinese from South China; JPT Japanese from Tokyo, Japan; CDX Chinese Dai from Xishuangbanna, China; KHV Kinh from Ho Chi Minh City, Vietnam; CEU Utah residents with Northern and Western European ancestry CEPH collection; GBR British from England and Scotland; IBS Iberian populations in Spain; TSI Tuscans in Italy; FIN Finnish in Finland; GIH Gujarati Indians from Houston, Texas, United States; PJL Punjabi from Lahore, Pakistan; BEB Bengali in Bangladesh; ITU Indian Telugu from the UK; STU Sri Lankan Tamil from the UK.

indicating that the Denisovan-like ancestry in the European populations was derived from some indirect gene flow due to recent admixture with other populations. The ancestral source populations were most likely some South Asian groups, as we observed similar levels of ancestry-sharing between European and South Asian populations (ancestry-sharing ratio: on

average 51.16, with a minimum of 40.97 and a maximum of 65.40) compared to ancestry-sharing of South Asian populations (ancestry-sharing ratio: on average 48.59, with a minimum of 44.58 and maximum of 61.10) (Supplementary Note 5.3). For comparison, the ancestry-sharing ratio in European populations was much higher, with 146.48 on average

**Table 3 Introgression history inference for Ust'-Ishim.**

| Archaic hominins | n[a] | Support ratio | Time (proportion) | Bootstrapping 95% CI |
|---|---|---|---|---|
| Denisovan | 1 | 100 | 108.9 (0.04)[b] | 92.3–147.6 (~0.04) |
| Neanderthal | 2 | 94 | 59.8 (1.54) | 57.8–61.4 (1.41–1.57) |
| | | | 152.7 (0.07) | 95.6–204.1 (0.04–0.2) |

[a]n denotes the number of introgression events.
[b]Time is measured as thousands of years ago (kya). Numbers in the brackets are the estimated introgression proportion (%).

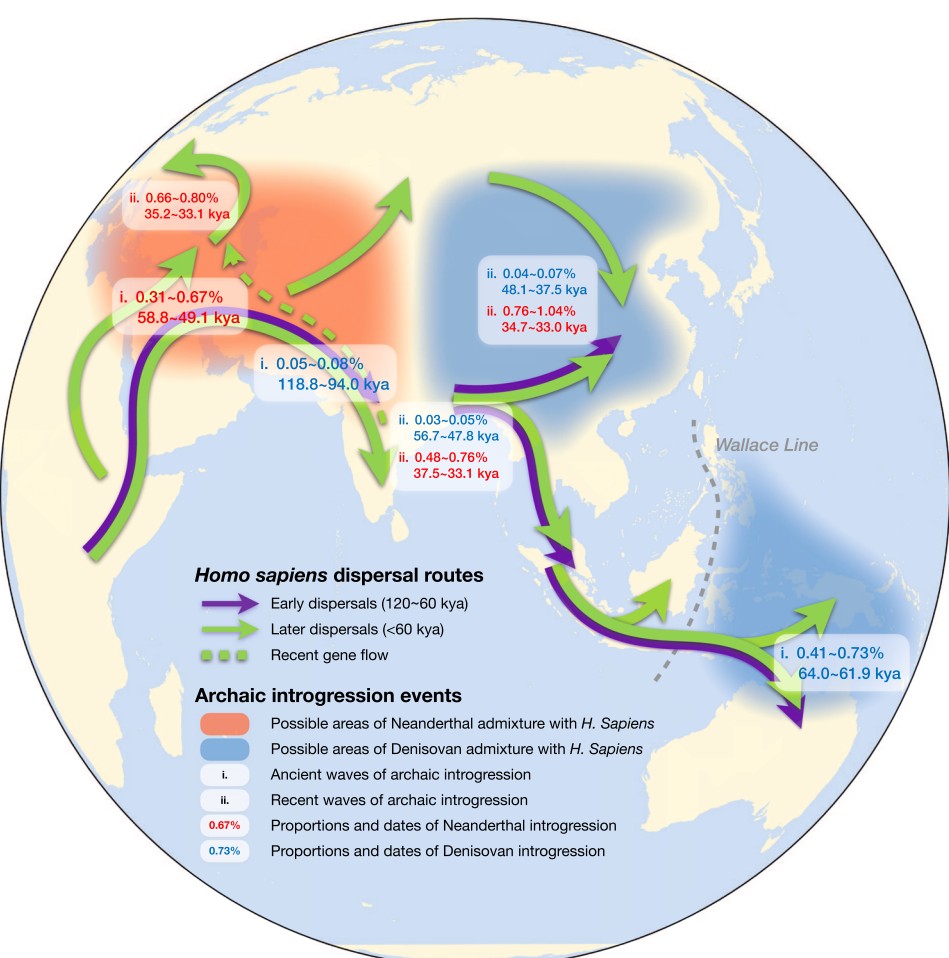

**Fig. 5 Map of *Homo sapiens* dispersal routes and admixture between archaic and modern humans.** Migration pathways of *H. sapiens* are supported by archeological evidence. Translucent red and blue represent possible ranges for contact between archaic and modern humans. The waves, admixture proportions, and dates inferred in our study are labeled in boxes in white. kya thousand years ago.

(a minimum of 122.34 and a maximum of 180.16) (Supplementary Note 5.3).

The introgression history model was consistent with the Denisovan-like ancestry-sharing ratio in Europeans. We detected ancient Denisovan-like introgression, which happened around 118.8–101.6 kya in all five KGP European populations. A weak recent Denisovan-like introgression was identified in these European populations, except for the most Western population, i.e., Iberian populations in Spain. The recent weak Denisovan-like introgression event can be explained as recent gene flow from some Asian populations of Denisovan-like ancestry.

**Reconstructing archaic–modern human admixture history.** Taken together, we detected two-wave Neanderthal-like introgression and two-wave Denisovan-like introgression in Eurasia.

The first-wave introgression from the two archaic hominins was shared among Eurasian populations, except for the Papuans, who experienced an independent Denisovan-like admixture different from that observed in populations west of the Wallace Line (the Papuans shared the two-wave Neanderthal-like admixture with populations west of the Wallace Line, but no detectable two-wave Denisovan-like admixture, as observed in populations west of the Wallace Line; the evidence is based on dating and ancestry-sharing patterns). The second-wave introgression from a Neanderthal-like group occurred independently in different regions (East Asia, South Asia, Europe), and the second-wave Denisovan-like introgression took place independently in East Asia and South Asia (but South Asia shared this wave with Europe). The European population underwent the second-wave Denisovan-like introgression via some recent gene flow from Asia (most likely from South Asia). There were two waves of

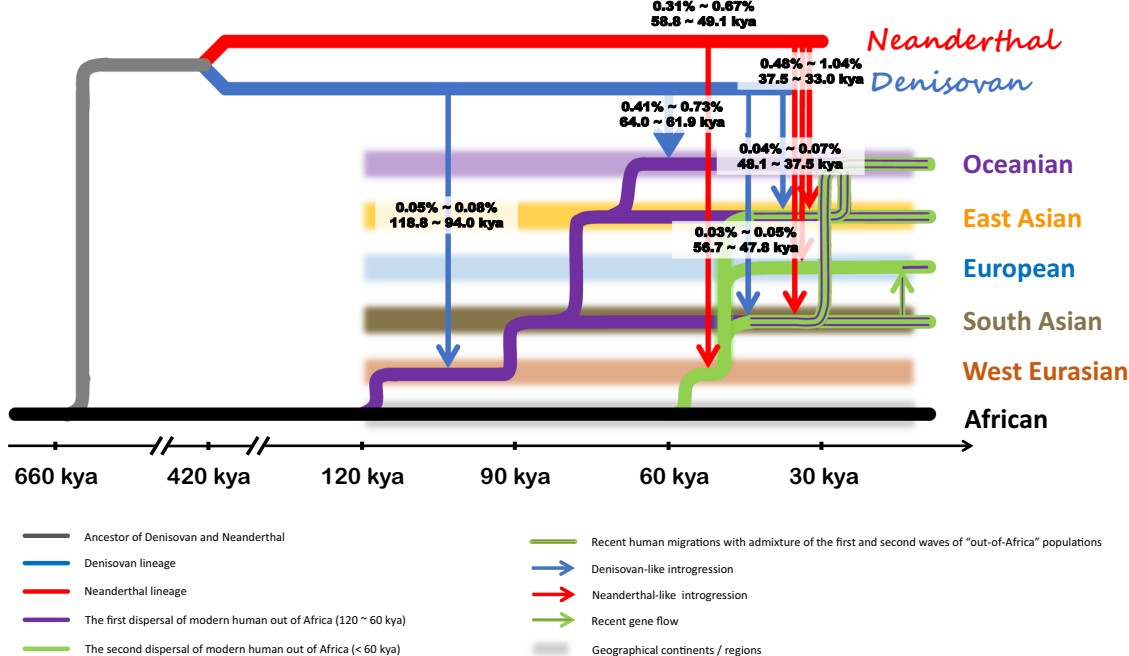

**Fig. 6 The landscape of prehistoric human dispersal and archaic introgression in Eurasia and Oceania .** The brown lines represent the lineages of the first-wave "Out of Africa" migration, the green line represents the second-wave "Out of Africa" migration, and the brown-green line represents the lineages admixed by the two-wave "Out of Africa" migration. Red arrow lines represent the Neanderthal-like introgression events and blue arrow lines represent the Denisovan-like introgression events. Shadow areas of different colors stand for different continents/regions.

Neanderthal-like introgression shared by Asians (East Asia, South Asia, or both) and the Papuans, but our results suggest that Neanderthal-like ancestry was likely via some indirect gene flow (Fig. 4), to which modern human migration to Oceania contributed (after the second-wave Neanderthal introgression occurred in places West of the Wallace Line, 37.5–33.0 kya; Table 2 and Figs. 5 and 6). The migration of modern humans to Oceania also brought Denisovan-like ancestry (which is not observable in the Papuans despite the obvious possibility, because of the migration history of the modern Eurasians to Oceania). However, an independent Denisovan-like introgression occurred in Oceania, other than the non-observed Denisovan-like ancestry brought by the modern human migration. Therefore, in principle, totally there could have been three-wave introgression of Denisovan-like DNA in the Papuan genomes, with one of them being dominant and different from that in any other Eurasian populations, which occurred 64.0–61.9 kya (Table 2).

The results in this study could not be explained by a single-wave "Out-of-Africa" model; in particular, the independent Denisovan-like introgression in Papuans occurred much earlier (64.0–61.9 kya) than both admixtures of Neanderthal-like DNA in the Papuan populations (first wave 61.7–53.0 kya, second wave 35.2–28.9 kya). It was also before the first-wave Neanderthal-like introgression in East Asia (56.9–49.1 kya) and that in South Asia (58.2–53.2 kya). We proposed a two-wave "Out-of-Africa" model to reconstruct the archaic–modern admixture history (Figs. 5 and 6). The first wave out of Africa happened around 120 kya[40]. We estimated that the first Denisovan-like introgression event occurred 118.8–94.0 kya, which indicated there might have been an archaic–modern admixture that happened around 120–80 kya in the Middle East. Skhul and Qafzeh hominins in Israel, which died around 120–80 kya, exhibit a mix of traits found in archaic and AMHs[41]. Then, the first wave of modern human ancestors migrated to Asia and Oceania. Madjedbebe, the oldest known site showing the presence of humans in Australia, suggests humans have been in these areas since at least 60–50 kya[42], which is much

closer to the time of Denisovan-like introgression in Oceania (64.0–61.9 kya). The second wave of modern humans went out from Africa around 60 kya to the Middle East. During that period, a branch of Neanderthals who lived in Amud Cave, Levant (60–50 kya)[43] might have encountered our ancestors. This also corresponded to the first wave of Neanderthal-like introgression into Eurasian genomes (58.8–49.1 kya). The second-wave modern humans moved worldwide through the Middle East and may have encountered Altai Neanderthals[2], European Neanderthals[1,3], Altai Denisovan[2], or some other undiscovered archaic hominins in South Asia (Figs. 5 and 6).

**Functional and phenotypic effects of archaic sequences**. To annotate and understand the effects of introgressed sequences, we leveraged association studies and prior biological findings in the literature and public databases. We identified 80 high-frequency (>0.3) Neanderthal-like segments shared by all Eurasian populations studied. Seven genes overlap with these regions, and interesting examples include *ZNF169*, involved in the immune-related pathways and associated with body mass index, and *HHAT*, associated with cardiovascular disease and lung function (Supplementary Data 1). We also identified discrepant immune-related genes with Neanderthal ancestry (frequency > 0.3) that are specified in different regional populations, including *NLRC5* in East Asians, *ATP1B1* in Europeans, *CCR1/CCR3* in South Asians, and *BCL2* in Papuans (Supplementary Data 2). Intriguingly, the high-frequency Neanderthal-like segments specific in East Asians showed significant enrichment in response to UV, mostly attributed to *HYAL1/HYAL2/HYAL3*, which encode for lysosomal hyaluronidases and also function in carbohydrate metabolism. A related gene in Europeans is *TCF7L1*, involved in melanogenesis. Papuans had larger amounts of specific Neanderthal-like segments than the other major continental groups, including genes involved in neurological functions, such as *DAB1* and *SLIT2*. We did not identify high-frequency segments shared by all Eurasian populations with Denisovan

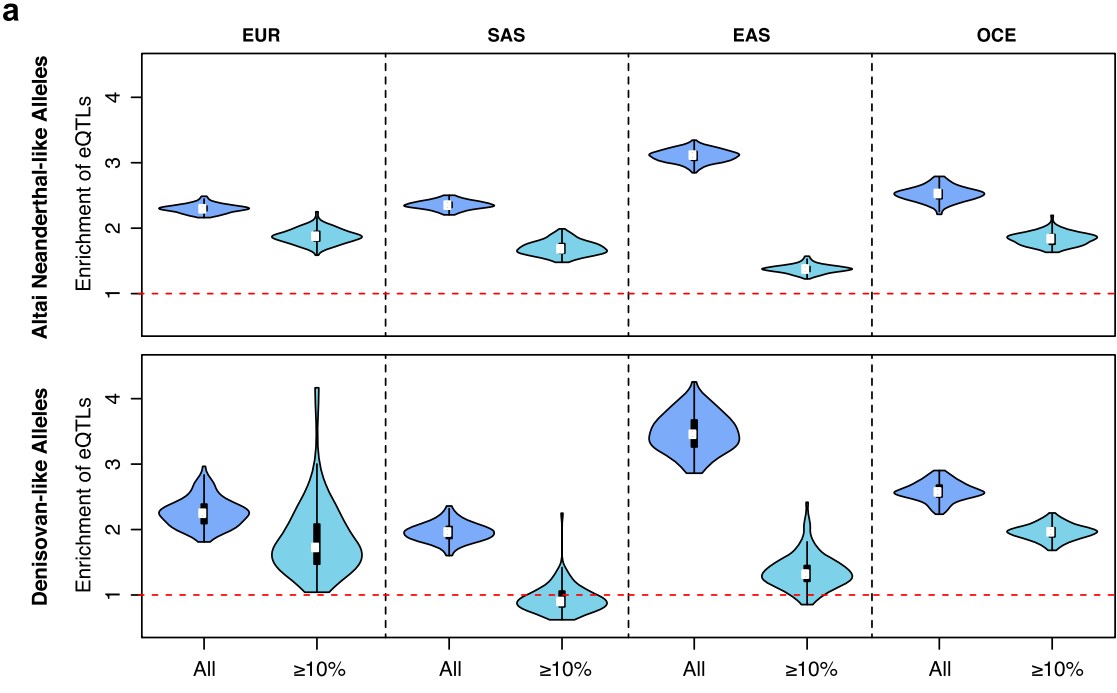

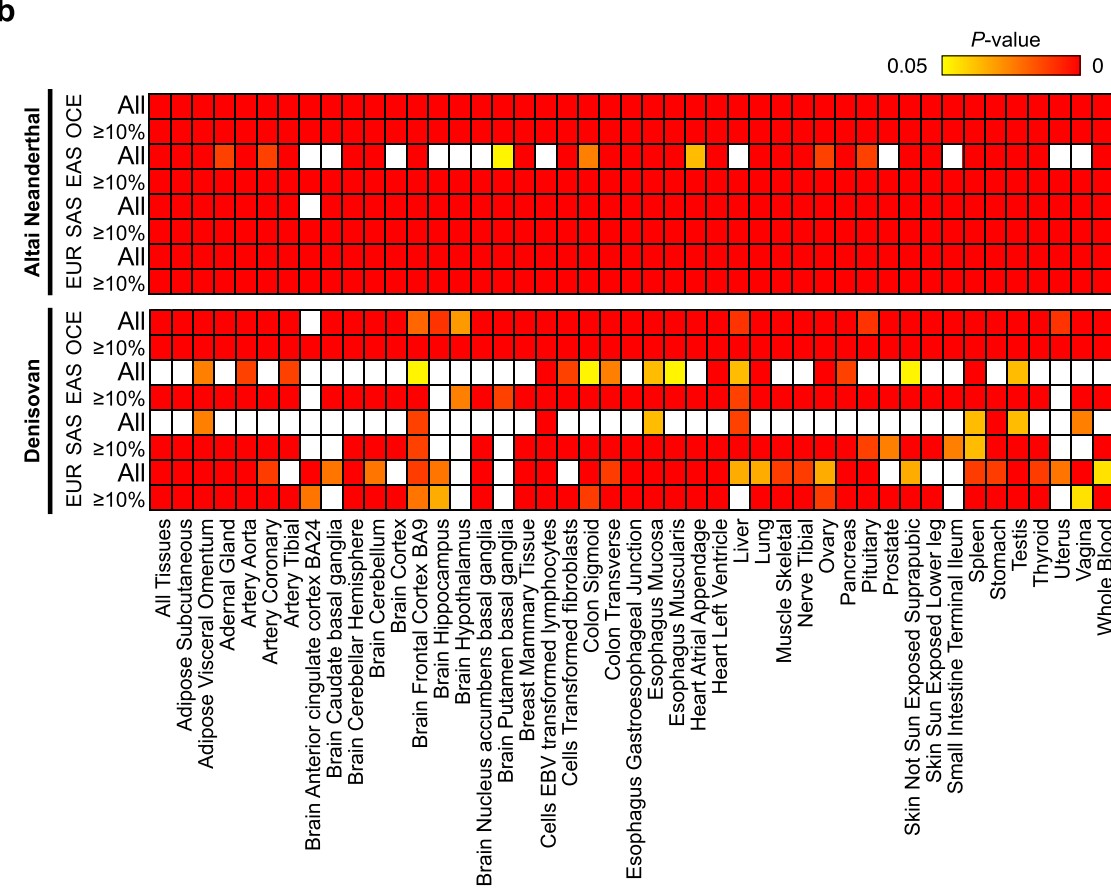

ancestry. Considering the distinct wave of Denisovan-like introgression in Papuans, we paid special attention to the Papuan-specific high-frequency Denisovan-like segments. These segments contain some immune-related genes (e.g., *ERBB2* and *IL7R*) and, interestingly, several photoreception-related genes, such as *ABCA4* (encoding ATP-binding cassette transporters and expressed exclusively in retinal photoreceptor cells) and *GJA10*

(coding for connexins involved in tracer-coupling between horizontal cells of the retina) (Supplementary Data 3).

In addition, a number of well-recognized genes with archaic ancestry have been replicated in our analyses. For instance, *BNC2* and *OCA2* (associated with skin pigmentation levels in non-African populations) and *OAS1/2/3* and *TLR1/6/10* (related to immunological functions) were identified to have Neanderthal

**Fig. 7 Enrichment of the archaic-like alleles in variants associated with gene expression regulation. a** Enrichment of Neanderthal-like (upper panel) and Denisovan-like (lower panel) alleles in the pooled eQTLs of all tissues. Each violin plot shows the ratio between the number of eQTLs in the archaic loci and that in the frequency-matched non-archaic loci. Two subsets of loci were analyzed, including all archaic loci (dark blue) and those with archaic allele frequency ≥10% (light blue). The white dot and the black bar in the center of each violin indicate median and interquartile range of the ratio, respectively. Whiskers are represented in the form of Tukey style. The red dashed line indicates ratio = 1. EUR European, SAS South Asian, EAS East Asian, OCE Oceanian. **b** Enrichment of the archaic-like alleles for eQTLs in each tissue. The Neanderthal-like alleles (upper panel) and the Denisovan-like alleles (lower panel) were analyzed independently. For each population group, two subsets of loci were analyzed, including all archaic loci and those with archaic allele frequency ≥10%. The yellow-to-red heatmap shows significant $P$ values ($P$ value <0.05 obtained by a one-sided empirical test and corrected using the Benjamini–Hochberg procedure accounting for all tissues; Exact $P$-values are shown in Supplementary Data 5), which means the proportions of eQTLs in the archaic-like alleles are significantly larger than those in the non-archaic-like alleles. In both plots, the EUR group consists of 503 European samples from CEU, FIN, GBR, IBS, and TSI; The SAS group includes 403 South Asian samples from GIH, ITU, STU, and PJL; The EAS group integrates 405 East Asian samples from CHB, CHS, JPT, and CDX; The OCE group is represented by 30 Papuan samples.

ancestry in our data (the maximum frequency of the Neanderthal-like haplotype of *BNC2* is 0.55 and 0.11 in Europeans and South Asians, respectively; the maximum frequency of the Neanderthal-like haplotype at *OCA2* is 0.2, 0.27, 0.64, and 0.7 in Europeans, South Asians, East Asians, and Papuans, respectively; the maximum frequency of the Neanderthal-like haplotype at *OAS1/2/3* is 0.33, 0.25, and 0.33 in Europeans, South Asians, and Papuans, respectively; the maximum frequency of the Neanderthal-like haplotype at *TLR1/6/10* is 0.22, 0.15, and 0.4 in Europeans, South Asians, and East Asians, respectively)[44,45]. Examples of certain populations carrying putatively Denisovan-like adaptive sequences are rather limited, except in the Oceanic populations. Vernot et al.[16] reported 21 adaptive introgression regions in Melanesians and 18 of them were also identified in Papuans using *ArchaicSeeker 2.0*, in which 15 showed high frequencies of archaic ancestry (>0.3; Supplementary Data 4). In these 15 regions, one is of Denisovan origin, encompassing *GALNT15*, involved in carbohydrate metabolism, and one distinguishes both Neanderthal and Denisovan haplotypes from modern human sequences but does not contain any protein-coding sequences. Others are of Neanderthal origin, including *GBP4* and *GBP7*, related to immune reactions, and *GCG*, responsible for blood glucose homeostasis. These findings provide clues to understand possible functional and phenotypic consequences of archaic introgression in modern humans. More efforts would be needed to draw the connections between the annotated biological functions of the archaic sequences and the original beneficial phenotypes.

Previous studies suggested that the phenotypic effects of archaic sequences are more likely mediated through gene regulation than through protein changes[45–47]. Therefore, we examined the enrichment of eQTLs in the archaic-like alleles based on the GTEx database (v6)[48]. For each population group, we compared the proportion of eQTLs in the archaic alleles with that in the randomly sampled frequency-matched non-archaic alleles (see "Methods"). Before the analysis, we clustered the SNPs with high LD ($r^2 > 0.8$), and the calculations were performed on approximately independent loci. We pooled the eQTLs from all tissues and observed a non-occasional enrichment of the eQTLs in the archaic alleles in all four population groups (Fig. 7a). Similar results were obtained after eliminating the low-frequency archaic-like alleles (<10%), except that the proportion of eQTLs in the Denisovan-like alleles did not show a substantial difference with that in the non-archaic alleles in South Asians. When testing each tissue, we found most of the tissues showed such significant enrichment, but the Denisovan-like alleles, in general, had less effect on the eQTLs than the Altai Neanderthal-like alleles (Fig. 7b and Supplementary Data 5). These results confirmed the conclusions of previous studies[47]. Although applying the eQTLs reported by the GTEx database to non-Europeans could cause biased results, the impact of archaic introgression on gene expression should not be ignored and deserves further investigation.

Despite the wide-spread archaic sequences across the genome of present-day human populations, we identified a set of archaic deserts showing depleted archaic ancestry, of which six extended up to 10 Mb in length (Supplementary Data 6). These deserts highly overlapped with those reported in the *S\** or *IBDmix* analyses[16,17,26,49] (Supplementary Data 7), including *FOXP2* (ref. [37]). Interestingly, the archaic deserts are significantly enriched in genes related to skin development and keratinization (Supplementary Note 5.4 and Supplementary Data 8), most of which (59 in 73) belong to the *KRT* (keratin) or *KRTAP* (keratin-associated protein) gene family. The underlying mechanisms of the archaic deserts are not yet fully understood, but some driving forces are expected to lead to the repeated loss of archaic ancestry at these regions across multiple independent admixture events.

## Discussion

Archaic human admixture has exerted a great influence on the genetic and phenotypic diversity of present-day human populations. However, detecting introgression and modeling admixture history remain challenging. In this study, we proposed a method called *ArchaicSeeker 2.0* which allows simultaneously detecting introgression sequences and inferring admixture history. Furthermore, we applied this method to refine the archaic human admixture history in Eurasia. To the best of our knowledge, this study is the first effort in modeling complex archaic human admixture history based on the length distribution of the archaic introgressed segments. *ArchaicSeeker 2.0* is a method to reconstruct the multiple-wave introgression into present-day human populations. Compared with other currently available methods[28,29] that allowed multiple-wave introgression scenarios, *ArchaicSeeker 2.0* not only reconstructs the introgression history, but also identifies the introgressed sequences in modern human genomes. The framework of our method is flexible, with known archaic hominin sequences as references, and this enables *ArchaicSeeker 2.0* to classify the candidate introgressed segments into ancestral catalogs more precisely and in a higher resolution compared with other methods. *ArchaicSeeker 2.0* is powerful to detect unknown archaic sequences; for example, in the simulation of scenarios of deep divergent and unknown archaic introgression, the TPR of our method is still greater than 81.9% and precision is above 90% (Supplementary Note 4.1.1). Nonetheless, our analysis also showed that the known archaic genomes were valuable in modeling ancient admixture of unidentified hominins. Moreover, *ArchaicSeeker 2.0* takes into account not only the tree topology but also the branch length of the phylogenetic tree (matching algorithm) and the length of the introgressed sequence to mitigate the influence of incomplete lineage sorting (Supplementary Note 2.4, 4.3).

We applied *ArchaicSeeker 2.0* to analyze global population data and reconstructed the archaic–modern human admixture history. We estimated the introgression proportion in different populations worldwide. Previous studies reported around 1–4% of Denisovan-like ancestry in the Papuan population[50,51] and around 1% in other Asian populations[4,35]. However, our estimation of Denisovan-like ancestry was much lower, i.e., ~0.61% in Papuan and ~0.1% in other Eurasian populations. We were aware that the proportion of archaic sequences detected in a present-day population could be strongly dependent on the type of methods used. Most site-based methods, such as *D*-statistics and *F*-statistics, typically produce a higher introgression estimation. For instance, Reich et al.[50] used *D*-statistics and estimated Denisovan contributed 4–6% of its genetic material to the genomes of present-day Melanesians, and Benjamin Vernot et al.[16] used an *f4* statistic and found 1.9–3.4% of Denisovan ancestry in Melanesian samples. In contrast, sequence-based methods often provided a much lower estimation. Taking the Papuan as an example, Skov et al.[18] reported a summation of the lengths of Denisovan-like introgressed sequences detected by different methods, i.e., 83.11 Mb by HMM[18], 43.11 Mb by Sstar[16], 58.17 Mb by CRF[17], and 38.98 Mb by Sprime[13]. These results are largely comparable to ours, i.e., 43.8 Mb in the Papuan. Rogers and Bohlender[52] showed that the estimators used in previous studies exhibit strong biases in the case of more than one source of archaic admixture. Their Fig. 4 (ref. [52]) showed that these estimates could be biased upwards by 600%, which was also recently confirmed by Rogers[28]. In our simulation studies (Supplementary Note 4.3), there were also lines of evidence indicating overestimation of the archaic introgression by the *D*-statistics for 5–10-folds.

Previous studies suggested that there is little evidence of Denisovan gene flow into Europeans[14,35]. However, we did identify a considerable proportion of Denisovan-like sequences in the present-day European populations. The relatively lower proportion and within-population diversity suggested that these Denisovan-like sequences could have resulted from recent gene flow from some Asian populations in an indirect way. Indeed, quite a few studies have provided solid evidence of gene flow from Asian to European populations[39,53]. These results also suggest that our method is suitable to identify archaic ancestry that is present at low proportions.

The new method enabled us to refine the estimation of archaic ancestry in present-day human populations as well as the models of archaic–modern human admixture in Eurasia. The two-wave admixture model provided new insight into modern human migration history in Eurasia. The extent of variation in archaic ancestry within and among Eurasian populations cannot be adequately explained by a single "Out of Africa" dispersal model. In particular, we estimated that the first wave of Denisovan-like introgression occurred in Eurasian populations around 118.8–94.0 kya, which was much earlier than 60 kya, when the initial peopling of Asia by modern humans occurred assuming a single "Out of Africa" dispersal model. Our analysis suggests modern humans arrived in Eurasia before 60 kya, indicating at least two waves of migrations occurred in the "Out of Africa" history of modern humans. Moreover, we found that the archaic introgression model in the Papuan populations was different from that in the Eurasian populations. Strikingly, despite a shared history of Neanderthal-like introgression between the Papuan and Eurasian populations, Denisovan-like introgression in Papuan was different from that in Eurasian populations. In particular, Denisovan-like introgression could be earlier than the first wave of Neanderthal-like introgression in Papuans. The Denisovan-like introgression in the Papuan population independently occurred eastern to the Wallace Line. Suppose there was only one wave of modern human migration to Oceania, the following two waves of Neanderthal-like introgression in Papuan would be expected to occur somewhere east of the Wallace Line, which should be independent of Neanderthal-like introgression in other populations. However, we did observe shared Neanderthal-like ancestry between Papuans and populations west of the Wallace Line. Therefore, these results could not be explained by a one-wave model of modern human migration to Oceania. A more plausible explanation would be that the populations eastern to the Wallace Line experienced at least two pulses of ancient admixture with modern human ancestors.

Several recent studies showed that there might have been gene flow from Neanderthals to African populations[49,54,55]. It is also plausible that the ancestors of Neanderthals and Denisovans interbred with some superarchaics[30]. Furthermore, previous studies also suggested that the sequenced Neanderthal and Denisovan received gene flow from modern humans[2,4]. The potential admixture history of archaic and African references used in the analysis could affect the power of detecting introgression sequences and result in underestimation of the admixture proportion. However, since the estimation of the AMH ancestry was only slightly affected (Supplementary Note 2.6), the overall picture of the population admixture history would not be changed much, as inferred and presented in this study.

## Methods

*ArchaicSeeker* is a series of software for detecting archaic introgression sequences and reconstructing introgression history. The latest version of this series, *ArchaicSeeker 2.0*, has the following three notable improvements compared with the original version of this software[23]. First, it can automatically determine the boundary of each introgressed sequence. Next, it is capable of tracing both known and unknown ancestral sources of a given introgressed sequence. Finally, it can be used to reconstruct the introgression history with more sophisticated introgression models (Supplementary Note 2). *ArchaicSeeker 2.0* consists of three modules, namely, seeking introgressed sequences, matching segments to proper ancestries, and reconstructing introgression history.

For the detection of introgressed sequences, our method utilizes both information from the population level and information from the individual/haplotype level. To this end, we used an HMM to describe the mosaic genome admixed by archaic hominins and modern humans and a modified expectation–maximization (EM) algorithm to estimate part of the HMM parameters. The accuracy of our method relies on the accuracy of both SNP level information and introgressed length information. Moreover, a likelihood-based matching method was built in our software to find the proper ancestries of each candidate archaic sequence. This matching method is flexible and powerful, allowing each candidate archaic sequence to match any ancestral lineage regardless of whether the ancestry is known or unknown. For introgression history inference, inspired by the methods in the *MultiWaver* series[31–33], we applied the General Discrete Admixture Model[31,32] to describe the multiple-wave introgression history. This model used the archaic tract length distribution to reconstruct the introgression history, which inferred the number of admixture waves of each ancestry with a likelihood ratio test (LRT), and used an EM algorithm to estimate the admixture time and the corresponding proportion of each wave.

**Seeking algorithm.** We used an HMM to describe the mosaic genomes admixed by archaic hominins and modern humans. The model has two hidden states: one represents archaic ancestry and the other AMH ancestry. Each SNP on the test genome is classified into one of the following four observation states. The first three states concern a test haplotype carrying the derived allele at one site. First, if all of the archaic and African references were ancestral alleles, we defined the observed state of this test haplotype at this site as State 1 (Test Pop-Specific Markers). Next, if the derived allele could be found in archaic references, but not in African references, this site was called State 2 (Archaic Markers). If the derived allele was shared between the test haplotype and African references, while it could not be found in the archaic references, this site was defined as State 3 (AMH Markers). All of the others were classified as State 4 (Common Markers).

For Hidden State 1 (archaic), the initial probability of that state was set to be the introgression proportion $\alpha$, and the initial probability of Hidden State 0 (AMH) was set to $1 - \alpha$. The initial value of $\alpha$ was set as 0.02 and introgression time was set as 2000 (generations) as the default. Our simulation results showed that choices of these initial values did not have much impact on the performance of our method. We calculated the transition probability matrix with introgression time and introgression proportion, where the transition probability between any two adjacent SNPs could vary according to the different genetic distances between them (Supplementary Equation (8)). The initial values of the emission probability matrix were calculated with a parameter $\varepsilon$, whose default value was set to 0.99.

Part of the parameters in the HMM was updated with an EM algorithm as follows. During each iteration, the introgression proportion and time were updated and then used to calculate the transition probability matrix. For the emission probability matrix, only six entries were updated each iteration while the other two entries were used as the initial value for reducing the false detection rate (Supplementary Notes 2.2 and 2.5).

**Matching algorithm.** This matching algorithm is a likelihood-based method that requires a prior matching model, which is the phylogenetic relationship of input populations, as the input. Before the matching algorithm is invoked, a model calibration step is performed to mitigate the uncertainty or bias of the prior model. The application step uses pairwise genomic differences, and a chimpanzee reference genome was introduced to the calibration step to set the root of the phylogenetic tree (Supplementary Note 2.1).

Based on the calibrated model, the algorithm matches each candidate archaic sequence to a proper ancestry, i.e., an edge on the matching model. The candidate ancestries included both known ancestries, which stood for ancestries present in the reference panel, and unknown ancestries, which represented the ancestors of some references. In addition to finding the best-matched ancestry, our method also estimates a split/divergence time to the reference populations for each candidate segment (Supplementary Note 2.3).

**Introgression history reconstruction.** In this model, we modified the General Discrete Admixture Model in *MultiWaver* software[31–33] to infer multiple-wave introgression events by setting the number of admixture waves from modern humans as one (Supplementary Note 2.6). In this model, we applied an LRT to select a proper number of introgression waves and used an EM algorithm to estimate the corresponding parameters (Supplementary Note 2.6).

**Archaic ancestry-sharing between populations.** Since only a few introgression events happened in history and the contribution from each single introgression event is expected to be small, the introgressed segments derived from the same archaic lineages at the same genomic position were likely inherited from a common archaic ancestor.

To determine the genomic position of introgression in any two given populations, we introduced a statistic named ancestry-sharing ratio of any two populations, $S_{ij}$, given as

$$S_{ij} = \frac{\sum_{k=1}^{n} \left( p_{ik} \times p_{jk} \times L_k \right)}{P_i \times P_j \times L}, \tag{1}$$

where $P_i$ stands for the genome-wide introgression proportion of population $i$ and $p_{ik}$ is the local introgression proportion of a genomic segment $k$ in population $i$. We assume there are $n$ segments in the genome and the local introgression proportion of the two populations is identical at any position in one segment. Each segment length is $L_k$ and $L = \sum_{k=1}^{n} L_k$ is the total length of the genome.

Intuitively, the statistic $S_{ij}$ measures the ratio of introgression-sharing to the random introgression-sharing of any two populations. Since the introgression proportions of different continental populations are different, two populations with higher introgression proportions tend to share a few more introgressed segments. We used random sharing to mitigate this potential effect and control excessive proportions of introgression. Detailed properties of statistics for archaic ancestry-sharing can be found in Supplementary Note 5.3.

**Simulations.** Massive simulation genomic data were generated by *ms*[56] to examine the performance of our software. We simulated introgression from both a single archaic lineage (see details in Supplementary Note 3.1) and two different lineages (Supplementary Note 3.2). In addition, scenarios with a deep divergent unknown archaic lineage were simulated to test the unknown archaic introgression detective ability (Supplementary Note 3.3). A script named *SimAncestry* was developed to determine the ancestral segments by analyzing the local tree topology based on *ms* output.

**Archaic allele identification.** Archaic alleles originated in the archaic lineages and entered modern human genomes via archaic introgression. An archaic allele is assumed to be a derived allele that is present in the introgressed sequences, but not in the genomic segments of modern human ancestry. Derived alleles on inferred archaic segments are very likely archaic alleles, while derived alleles which can be observed in African populations are less likely archaic alleles. Two factors influence the probability that such a derived allele is an archaic allele. One is the probability that this allele is present on an inferred modern human sequence ($p_d^M$), and the other is the probability that the allele is present in the African genome ($p_d^{Afr}$). We can easily estimate $p_d^M$ and $p_d^{Afr}$ through the observed derived allele frequencies in the inferred modern human and African genomes. Let $A$ be the random event where the derived allele on an inferred archaic segment is an archaic allele. The probability $P(A)$ can be estimated based on these two probabilities.

First, if a derived allele on an inferred archaic segment is an archaic allele, ideally, $p_d^M$ is expected to be 0. However, due to limitations of the detection power, the archaic allele may also occur in the inferred modern human genome. Let $B$ be the event where this derived allele is present on inferred modern human segments. We note that $P(B)$ is equal to $p_d^M$. Assuming the power of our method to detect archaic segments is $p$, the probability that the derived allele is archaic is

$$P_1(A) = P(B)P(A|B) + P(\bar{B})P(A|\bar{B}) = p_d^M(1-p) + (1 - p_d^M). \tag{2}$$

Second, if the derived allele is archaic, $p_d^{Afr}$ is also expected to be 0, ideally. However, there might be back-wave gene flow from Eurasian to African populations, so the derived allele may also occur in African genomes. Let $C$ be the event where this derived allele occurred on African genomes and $P(C) = p_d^{Afr}$. Suppose the admixture proportion is $m$ and the archaic ancestry proportion in Eurasians is $\alpha$. Then,

$$P_2(A) = P(C)P(A|C) + P(\bar{C})P(A|\bar{C}) = p_d^{Afr} \times m \times \alpha + (1 - p_d^{Afr}). \tag{3}$$

In summary, based on the frequency of the derived allele in the inferred modern human genomes and African genomes, we finally have

$$P(A) = P_1(A) P_2(A) = \left[ p_d^M(1-p) + (1 - p_d^M) \right] \left[ p_d^{Afr} m\alpha + (1 - p_d^{Afr}) \right], \tag{4}$$

$$P(\bar{A}) = 1 - \left[ p_d^M(1-p) + (1 - p_d^M) \right] \left[ p_d^{Afr} m\alpha + (1 - p_d^{Afr}) \right]. \tag{5}$$

If the probability that the derived allele on an archaic segment is not archaic $P(\bar{A})$, is less than 0.05, we regard this derived allele as an archaic allele, because this event is of low probability.

**Functional annotation and enrichment of the archaic sequences.** Genes were mapped to the putative introgressed genomic regions according to the Ensembl database version 96 using the GRCh37 coordinates[57]. Functional annotation and gene enrichment analyses were performed using *clusterProfiler*[58] implemented in R version 3.5.1 (ref. [59]), based on disease ontology, gene ontology, and KEGG pathway analysis.

To test the enrichment of the eQTLs in the archaic alleles, we first estimated LD for SNPs within a 500-kb sliding window using *PLINK v1.9* (ref. [60]), and then clustered the genome-wide SNPs with high LD ($r^2 > 0.8$) in each population group. Each of the SNPs that could not be tagged by any others represented a single cluster. We further defined one cluster as an archaic locus when at least one archaic allele was in this cluster, or as a non-archaic locus if no archaic allele existed. The archaic/non-archaic loci could be further determined as archaic/non-archaic gene expression-associated (GEA) loci if at least one of the linked SNPs was reported as an eQTL in the GTEx database (v6). Then we calculated the proportion of GEA loci in the total archaic loci. To make a comparison, we randomly selected 100 sets of non-archaic loci with equal numbers and matched frequencies of the archaic loci, and calculated the enrichment score as $\frac{\# \text{ archaic GEA loci}/\# \text{ archaic loci}}{\# \text{ non-archaic GEA loci}/\# \text{ non-archaic loci}}$. Enrichment score >1 indicates significant enrichment of the eQTLs in the archaic loci. Empirical $P$ values were obtained using a one-sided empirical test, and were then corrected using the Benjamini–Hochberg procedure accounting for all tissues.

To match the frequencies of the archaic and non-archaic loci, we selected a representative allele for each locus. For archaic loci, it was ideally an archaic eQTL in the archaic GEA locus or a randomly selected archaic allele in the archaic non-GEA locus. However, we could not always find an archaic eQTL in the archaic GEA locus; then we preferred to choose a random archaic allele as the representative allele. For each non-archaic locus, we selected a random eQTL in the GEA locus or a random allele in the non-GEA locus.

**Reporting summary.** Further information on research design is available in the Nature Research Reporting Summary linked to this article.

## Data availability

The KGP dataset used in this study is available at [http://ftp.1000genomes.ebi.ac.uk/vol1/ftp/release/20130502/]. The EGDP dataset used in this study is available under accession code PRJEB12437. The SGDP dataset used in this study is available under accession code PRJEB9586. The Altai Denisovan data used in this study are available at [http://cdna.eva.mpg.de/neandertal/altai/]. The Altai Neanderthal data used in this study are available under accession code ERP002097. The Siberian Ust'-Ishim data used in this study are available at [http://cdna.eva.mpg.de/ust-ishim/]

## Code availability

*ArchaicSeeker 2.0* was implemented with the programming language C++. The source code and user manual of *ArchaicSeeker 2.0* are freely available at https://github.com/Shuhua-Group/, https://www.picb.ac.cn/PGG/, and https://doi.org/10.5281/zenodo.526693.

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

## Acknowledgements

The authors thank Dr. Richard Durbin and Dr. Li Jin for their helpful discussions. This study was supported by the Strategic Priority Research Program (XDPB17, XDB38000000) of the Chinese Academy of Sciences (CAS), the National Natural Science Foundation of China (NSFC) grant (32030020, 32041008, 31771388, 91731303, 31525014, 31961130380, 31900418, 31871256, and 11801027), the National Key Research and Development Program (2016YFC0906403), the UK Royal Society-Newton Advanced Fellowship (NAF\R1\191094), Science and Technology Commission of Shanghai Municipality (19YF1455200), the Fundamental Research Funds for the Central Universities (2020RC001) and the Shanghai Municipal Science and Technology Major Project (2017SHZDZX01). The funders had no role in study design, data collection, and analysis, decision to publish, or preparation of the manuscript.

## Author contributions

S.X. conceived and designed the study and supervised the project. K.Y., X.N., C.L., and R.Z. developed the algorithm. K.Y. and R.Z. wrote the computer code. R.Z. contributed to computer simulation and testing code components. L.D. and X.M. performed natural selection analyses. K.Y., X.N., C.L., Y.P., Y.G., X.G., and J.L. performed statistical, mathematical, and computational analysis. K.Y., X.N., and L.D. wrote the original draft of the paper. S.X., T.W., and H.L. revised the paper. All authors reviewed and approved the manuscript.

## Competing interests

The authors declare no competing interests.
