## [Peer Review File · Nature Communications]

Refining models of archaic–modern human admixture in Eurasia with ArchaicSeeker 2.0Reviewers' Comments:

Reviewer #1:

Remarks to the Author:

Yuan et al extended a previously published algorithm to identify introgressed fragments and to estimate the time of introgression. The method is based on a Hidden Markov Model after applying a preprocessing step (definition of the phylogeny) and a post-processing (matching algorithm). The model for archaic introgression considered in this manuscript is the classical one, where the African population is not sharing archaic introgression and there are two archaic populations (Denisovans and Neanderthals). The authors benchmarked the method in a set of demographic models that include multiple introgressions. When applied to real data, the authors identify multiple introgression signals in all the considered populations and conclude given the estimated time of introgression that there has been more than one Out of Africa.

The methodology is interesting and some of the results are surprising. It can certainly change thinking in the field. However, I think that further evidence would be required to strengthen the conclusions. In particular, I am not so convinced by the benchmarking of the method and I think that the data that the authors have used (and how has been processed) could partially explain the observed results.

Main points

Low amount of Denisovan-like ancestry in Papuan (almost 10 fold) compared to other methods. The explanation provided by the authors is that the introgression was defined into specific hominin groups. Nevertheless, I think that this is not really explaining such proportions. If the authors add over all the introgressed regions from the different specific hominin groups, do they obtain a similar amount of introgression as with other methods?

Presence of Denisova ancestry in Europeans (line 544). The explanation suggested by the authors (line 548) is that this Denisovan ancestry is due to recent gene flow from some Asian populations in an indirect way. There are three points here that must be clarified. First of all, the evidence of recent gene flow between Europeans and Asians must be proven by means of algorithms such as ADMIXTURE or similar ones. To the best of my knowledge, such admixture has not been reported in the considered datasets. Second, the authors should show that the method is able to identify introgressed regions from recently admixed populations, which is not tested in their benchmarking. Third, if this Denisovan ancestry is due to recent admixture, the authors could check that Asian populations show the same signatures of introgression (i.e. in the same genomic regions as observed in Europeans). Finally, the authors could check the surrounding area of the Denisovan introgressed fragments and prove that they are of Asian ancestry.

The identification of old introgressions and their interpretation is interesting. However, I have some concerns about the validity of the data that the authors have used to reach these conclusions. First, the authors exclude Vindija because it uses a different calling method that provides a lower number of SNVs (however, it is used in the example of the Newick notation $((YRI:100,Test:100):557.5,(Denisovan:340,(Vindija33.19:82.5,Altai:20):280):237.5))$ (?). What would happen if the authors had used this genome? Is the method dependent on the SNV calling (I guess so, even if it is stated in S2.7 that it is robust to errors)? In that case, which kind of SNV calling pipeline is applied to ensure that the different datasets are called in the same way? Similarly, how is the deep coverage at each position taken into account when calling each SNV? This extends to the application of SHAPEIT2 to Ust'Ishm and the SGDP. It is known that SHAPEIT2 can produce artifacts when the reference populations (1000K) differ from the target populations (for example, Papuans or an ancient DNA sample). How were the non-called SNVs treated? I really hope they were not imputed by SHAPEIT2. Also, how was the derived state of each SNV estimated?

Furthermore, in supplementary materials it is stated "conducted a joint calling with another 1508

unpublished sequencing data generated by our group". Is this dataset used in this study? If not, which is the purpose of using it for the calling?

I think that the authors should first prove that the joint dataset that they use is not biased in one way or another. I would suggest checking that samples from the same geographic region (but from different datasets) cluster together independently of dataset in classical PCA/ADMIXTURE analyses, as well as that the results from their software are not dataset dependent (for example, for each geographic region check that the data from the different datasets provide similar results of introgression(not clear checking S5.1...)). In addition, have the authors assessed the power of the method against simulated datasets that include some of the scenarios for which the method should be robust (modern contamination, sequencing errors, etc)?

Another main point that I think the authors should clarify is the benchmarking:

1) The simulated datasets are constructed with ms. This software is not providing the introgressed regions and the authors implemented a script to infer the topology of the tree (SimAncestry) at a given genomic region (I guess, no information is provided about how this algorithm works). How is this script dealing with incomplete lineage sorting? How dependent is this inference to the SNV density in the genomic region? In other words: to which extent are the authors not biasing their conclusions by imputing the ancestry of these regions by means of SimAncestry? Or, alternatively, why not directly applying SimAncestry to the real data? In any case, I think it is important to provide a much more detailed explanation of this software, given that the benchmarking is basically done with its output.

2) The authors must provide the references for all the parameters that are used in the simulations. The time of the Altai Neanderthal seems to be much older than the one reported in the original publication (Prufer et al, ~50kya in contrast to the 60kya of the simulations and the same for Denisovans (in contrast to the 80kya)).

3) Related to the previous point, can the authors show that the genetic variation obtained in the simulations resemble the genetic variation observed in real populations? I am asking this question because the effective population sizes that the authors are providing do not match any of the described values in the literature (i.e. 1M chromosomes for Non African). I understand that these values are obtained after a phase of exponential growth, but in any case it is a substantial amount of chromosomes and I guess that this can influence the power for detecting introgressed fragments.

4) A bigger concern is the provided ms code. I understand (from the ms code) that the authors are simulating 10Mb each time. However, this information is not present (at least I have not been able to find it) in the description of the simulations. Following the ms code, I am not sure I understand the effective population size of Denisovans (3 and 4) and Neanderthals (5 and 6)(among other pops):
-n 3 1e-10 -n 4 1e-10 -n 5 1e-10 -n 6 1e-10 -n 7 1e-10 -n 8 1e-10
Similarly, I cannot find in the ms code that Denisovan and Altai Neanderthal are ancient samples (-eA option of ms)(?). Does this mean that they are sampled from the present in the simulations? Another problem is the growth rate: -eg 1e-10 1 175.328. In ms time is specified in terms of $4 * Ne_0$ generations. What does a scaled time of 1e-10 mean in the simulations (according to the SI, the time of the second expansion is $TAMH_{exp2}=5kya$ and of the first $TAMH_{exp1}=23kya$)? Similarly, a growth rate of 175.328 is extremely big. All this will increase the number of rare mutations, enhancing the possibility to distinguish between populations.

5) Generation time. This is not a parameter that one applies to ms and it should be specifically stated in the main text, in addition to the mutation rate and recombination rate (with a proper reference in all cases). Furthermore, despite there is no clear standard for the generation time, it is usually in the range of 29-30 years/generation using Hunter-Gatherers (i.e. Fenner 2005). This could be a potential problem for the estimates that the authors provide, based on a 25 years per generation (which has

also been claimed in other studies but at least the authors should cite them). Using 29 years/generation, a much older time estimates are obtained. However, this could be partially explained if other (higher) mutation rates were used.

Another point is that it is known that Denisovans and Neanderthals admixed, both between and also with other archaic populations. How would this affect the inference of introgression and time?

Other minor points:

1) What does "The introgression time and introgression proportion of each wave changed in gradient" mean?

2) I am not convinced about this sentence "Simulation results showed that ArchaicSeeker2.0 have a high precision (~93.0%), a high true positive rate (~90.4%) and a low false positive rate (~0.14%) in the archaic segments detection. In most simulations, more than 80% introgressed segments matched to the correct ancestry. For the introgression history inference, ArchaicSeeker2.0 inferred the correct scenarios for most cases (122/144, 84.7%)." How likely is that, if the authors distribute at random the fragments in the simulated 10Mb, they observe the same amount of overlap as with the inferred fragments?

3) A related question is if the percentage of inferred introgression in the genome correlates with the amount of introgression (in principle, the correlation should be quite high if there are no problems with the simulations).

4) S4.1 and S5.1.. What happens if the software is run with different hyperparameters (i.e. instead of considering introgression at 2% we run at 10%)?

5) S4.1.2. Unclear what extra information is providing the usage of SNPs since these are ultimately used for defining the introgressed fragments. For the Non-African AIMS, the statistics of the performance are computed considering only the Non-African AIMS or assuming all the SNPs?

6) "The total length test non-African genome of 90 repeats is $90 \times 200 \times 10 = 1.8 \times 10^{11} \text{bp}$ " in SI This sentence is wrong. I guess the authors mean $90 * 200 * 10^7$

7) It looks to me that many of the times of introgression are systematically underestimated or overestimated (STable 4.15).

8) It is unclear why an EM method is more "friendly and robust" to estimate the parameters. In my opinion, the EM algorithm should be quantified in terms of convergence towards the true value.

9) The paper claims that it is not model dependent, but it is obvious that it requires a basic demographic model to be fitted (non-admixed outgroup, relationship of the archaic populations).

10) The section of the Matching Model is confusing. For what I see, it is not taking the effective population size of the different populations into account to estimate the time of divergence.

11) The SI and Main text will require extensive English editing.

Reviewer #2:

Remarks to the Author:

The authors introduce a new method for detecting archaic introgression and dating the admixture times. The authors have performed a lot of tests and have done a good job of documenting them. But I fear its the wrong kind of tests and I am not convinced this method works as well as the authors

claim it does.

Major concerns

My main concern is that many of the admixture proportions and admixture times do not fit with literature.

For instance they detect a second wave of Neandertal admixture into European populations which occurred between 37.5-33.0 thousand years ago. All the evidence I have seen dates the last known Neandertals around 40 thousand years ago (Bard et al 2020, Higham et al 2014). The date of admixture is also substantially younger than what is estimated elsewhere (Moorjani 2016, Sankararaman 2016).

If the authors are correct this would be a groundbreaking claim - the authors need to comment on this!

Another instance is the amount of identified Denisovan sequence in Papuans is very low! The authors estimate around 0.41%-0.73%. From Dstatistics (3-6% Prufer et al 2014), other methods (>2% Denisovan Skov et al 2018). This suggests to me that the method is highly affected by how close the introgression denisovan is to the sequenced Denisovan. And if it's the case that the authors only pick up a fraction of all the Denisova sequence which is there (the part that is most similar to the sequenced Denisovan genome presumably), any claim about admixture timing becomes difficult.

I have looked at the simulation studies, because the results from this is the main argument that the method works well. Here I have some major concerns. The simulations are in a sense too well behaved. The population sizes are small (like the ancestral population size of modern humans is 3000 - but should probably be more like 7000 Gutenkunst et al 2009 - so things coalesce quickly), the recombination map is constant and it's assumed that one can call genotypes perfectly at every position in the Neandertal and Denisova genomes. In order to really assess their model the authors absolutely need to do the following things:

- 0) Use msprime instead of ms commands - it's much easier to use and maintain the code for.
- 1) increase effective population sizes (I would recommend the demography from Skov et al 2020 which is based on the Gutenkunst paper)
- 2) add varying recombination rate!
- 3) mask out SNPs in the archaics that fall outside the mappable regions (<https://bioinf.eva.mpg.de/map35/100/>)

Rerun all the analysis with this new demography and evaluate the method on these simulations instead. Before we know how well the method works on these more realistic simulations it does not make sense to evaluate any claims made about admixture times and admixture proportions.

Minor comments

Alien DNA?

I would call it ancient hominin/archaic DNA.

Ancient DNA

The authors does not include the Vindija genome because the number of SNPs is much less than the 2 previously published genomes - but there is no citation for this? How much less data? The Vindija Neandertal is much closer to the introgressing Neandertal than the Altai Neandertal and since the detection method relies on matching with archaic DNA I would think the segment determination would be more accurate using this genome.

Later the authors write:

In our method, we used African whole genome sequencing data (YRI) and archaic high coverage whole genome sequencing data (Altai Denisovan, Altai Neanderthal or Vindija33.19) as the references.

So are you using the Vindija genome or not?

Phasing of data.

For those SGDP populations which are not well represented by the 1000 genomes populations phasing is going to work less well. This will surely effect the how well you can determine archiac sequences?

April 5, 2021

Letter to Reviewers

RE: **NCOMMS-20-36650-T.R1**

Dear Reviewers,

On behalf of all the authors of this manuscript, we would like to thank you for your comments and suggestions which offered a great opportunity for a significant improvement of our manuscript entitled “***Refining models of archaic–modern human admixture in Eurasia***”.

We can see that both reviewers had spent much time assessing our manuscript, in particular, evaluating the method details. It was especially appreciable during these very difficult times of COVID19 pandemic. We truly appreciate all the constructive comments and suggestions from both reviewers. After studying your comments carefully, we understand that the main concerns are centered at the benchmarking of the method. Consequently, we conducted several additional analyses, and have carefully addressed each of the concerns from the two reviewers. We provide a point-to-point response to your comments and also an extra copy of the manuscript with all the changes highlighted. We hope our efforts made during the last two months would convince you and meet with your approval.

Sincerely yours,

Kai Yuan, Xumin Ni, Taoyang Wu & Shuhua Xu

Authors' Response to Reviewer #1 (Remarks to the Author):

Yuan et al extended a previously published algorithm to identify introgressed fragments and to estimate the time of introgression. The method is based on a Hidden Markov Model after applying a preprocessing step (definition of the phylogeny) and a postprocessing (matching algorithm). The model for archaic introgression considered in this manuscript

is the classical one, where the African population is not sharing archaic introgression and there are two archaic populations (Denisovans and Neanderthals).

The authors benchmarked the method in a set of demographic models that include multiple introgressions. When applied to real data, the authors identify multiple introgression signals in all the considered populations and conclude given the estimated time of introgression that there has been more than one Out of Africa.

The methodology is interesting and some of the results are surprising. It can certainly change thinking in the field. However, I think that further evidence would be required to strengthen the conclusions. In particular, I am not so convinced by the benchmarking of the method and I think that the data that the authors have used (and how has been processed) could partially explain the observed results.

Authors' Response: We thank the reviewer for the positive comments on our methodology, and we also understand the concern of the reviewers. Consequently, we spent more than two months completing the new benchmarking of our method following the suggestions of both reviewers.

Main points

Comment 1: Low amount of Denisovan-like ancestry in Papuan (almost 10 fold) compared to other methods. The explanation provided by the authors is that the introgression was defined into specific hominin groups. Nevertheless, I think that this is not really explaining such proportions. If the authors add over all the introgressed regions from the different specific hominin groups, do they obtain a similar amount of introgression as with other methods?

Authors' Response:

We thank the reviewer for this comment. We agree that the Denisovan-like introgression ancestry proportion in Papuan detected in this study is much lower compared with the results reported by some previous studies. To investigate whether such a low proportion is due to the matching algorithm, we counted the total candidate introgressed sequences, including Denisovan-like, Neanderthal-like, and some other ancestries, and found the total proportion is still less than 2%. Therefore, we agree with the reviewer that the overall estimation of the archaic ancestry based on our method is lower than those reported in

some previous studies, which cannot be simply explained by matching into different specific hominin groups. However, in our simulation, which based on the simulation scheme used by Vernot and Akey (Science 343(6174):1017-1021, 2014), there is no such significant underestimation (“almost 10 fold”) of the total archaic ancestry, neither did matching into different sub-lineages of the archaic groups (Table S4.13, S4.14).

Considering the reviewer’s concerns about our benchmarking (see Comment 9-14 below), we performed a series of simulations based on the model specified in Skov et al (Nature, 582:78-73, 2020), that is, assuming 8% Denisovan-like introgression and 2% Neanderthal introgression. We simulated 10 Mb sequences of 100 Africans (200 haplotypes), 100 Non-Africans (200 haplotypes), 1 Neanderthal (2 haplotypes) and 1 Denisovan (2 haplotypes) and repeated it 100 times. We inferred totally ~8.54% archaic introgressed sequences.

We further evaluated the accuracy of our results by comparing the correctly inferred lengths. The mean value of Precision is 94.72% (92.47% ~ 96.06%), TPR is 84.37% (80.45% ~ 88.01%) and FPR is 0.50% (0.32% ~ 0.76%) (Figure R1). The number interval in the brackets is the 95% CI. As you can see, there is some slight underestimation of the total archaic ancestry compared with that was set in the simulation (10%). Therefore, there is some slight underestimation of the archaic sequences by our method in these simulations, but definitely much less than “10 fold”.

Figure R1 Length based evaluation with Skov’s simulation scenarios. Comparison between the inferred introgressed segments and the ground truth segments under the scenarios described in Skov et al 2020. The y-axis represents the summary statistics of precision, TPR and FPR, respectively.

To figure out the root of the inconsistency between our estimation and that of some previous studies, we made a systematic investigation of the literature and studied the details of the methodology as well as the data type used. It turned out that most of the site-based methods which were originally designed for analyzing genotyping data, such as D-Statistics, F-Statistics make a higher introgression estimation, For example, David Reich et al. (Nature 468, 1053–1060, 2010) used D-Statistics and estimated Denisovan

4–6% of its genetic material to the genomes of present-day Melanesians. Benjamin Vernot et al. used the f_4 statistic and found significant evidence of Denisovan ancestry ($Z > 4$) in Melanesian samples, with admixture proportions varying between 1.9 and 3.4%.

On the contrary, sequence-based methods detected much less archaic ancestry compared with that estimated by the site-based methods. For instance, Skov et al (Table 1 in PLOS Genetics 14(9), e1007641, 2018) presented a summation of the lengths of Denisovan like introgressed sequences from different methods. Taking Papuan as an example, HMM-based method detected 83.11 Mb, Sstar detected 43.11 Mb, CRF detected 58.17 Mb, and Sprime detected 38.98 Mb Denisovan sequences. These results are comparable to ours, i.e., 43.8 Mb. The “seemingly” underestimation might due to the differences between different types of methods. Some degree of the difference also attributes to data types, i.e., early studies mainly analyzed genotyping data generated by SNP array, more recent studies had more chances to analyze deep-sequencing genomes. For example, Sundararajan et. al. (Table 1 in Curr Biol. 26(9):1241-7, 2016) reported just $0.85 \pm 0.43\%$ Denisovan introgression in Oceanian using the f_4 test.

Taken together, we are confident that the analysis based on our method provides a reliable estimation, while earlier studies might have overestimated the archaic ancestry in the present-day human genomes. We have updated our manuscript with these messages in the Discussion.

Comment 2: Presence of Denisova ancestry in Europeans (line 544). The explanation suggested by the authors (line 548) is that this Denisovan ancestry is due to recent gene flow from some Asian populations in an indirect way. There are three points here that must be clarified. First of all, the evidence of recent gene flow between Europeans and Asians must be proven by means of algorithms such as ADMIXTURE or similar ones. To the best of my knowledge, such admixture has not been reported in the considered datasets.

Second, the authors should show that the method is able to identify introgressed regions from recently admixed populations, which is not tested in their benchmarking.

Third, if this Denisovan ancestry is due to recent admixture, the authors could check that Asian populations show the same signatures of introgression (i.e. in the same genomic

regions as observed in Europeans). Finally, the authors could check the surrounding area of the Denisovan introgressed fragments and prove that they are of Asian ancestry.

Authors' Response: We thank the reviewer for these comments and constructive suggestions.

First, we should clarify that the “recent” admixture referred to in our manuscript is just relatively “recent” compared with the archaic introgression, it might be still hundreds of generations ago.

Second, we did manage to find some previous studies indicating recent gene flow between (South) Asians and Europeans. This includes the 1000 Genome Project paper by Auton, A. *et al.*, (Nature 526, 68–74 2015), where Figure 2a and Extended Data Figure 5 (ADMIXTURE results) show that there are certain shared genetic components between South Asians and Europeans. Furthermore, Qin et al. (Scientific Reports, 2015) quantified and dated recent gene flow between European and East Asian and explicitly demonstrated substantial gene flow from Asian to European populations. Lazaridis I et. al. (Nature 513, 409–413, 2014) reported three ancestral populations for present-day Europeans: One of them came from a ‘basal Eurasian’, who might be related to an early settlement of the Levant or Arabia, suggesting the substantial gene flow from Asia to Europe.

Next, following the suggestion of the reviewer, we did some additional benchmarking work by testing recent admixture with introgressed archaic sequences, and we showed that our method is able to identify the introgressed archaic sequences introduced by a recent population admixture. We simulated populations admixed 20-80 years ago and found there is no loss of TPR or Precision, nor is FPR increasing. In fact, the TPRs are around 90% and precisions are above 94%, while FPRs are only ~0.25% (see Figure R2).

Figure R2 Length based evaluation with scenarios of different admixture time. Comparison between the inferred introgressed segments and the ground truth segments under the recent admixture scenarios. The x-axis represents the admixture time and the y-axis represents the summary statistics of precision, TPR and FPR.

Furthermore, we searched the Asian genomes for the European signature of archaic introgression and managed to identify some notable examples as shown in Table R1.

Table R1 Denisovan-like Sequences Shared between European and South Asian

No.	Chr	Start	End	Length	SNVs	TagSNV						
1	1	41507431	41728268	220837	4526	12	103.097	112.635	3	6	5	
2	2	184460078	184585972	125894	1242	6	39.423	39.197	6	6	7	
3	2	228286321	228434783	148462	4087	49	110.251	122.168	3	8	10	
4	3	189918726	190021588	102862	2828	6	75.547	96.062	4	7	7	
5	7	49513215	49617216	104001	2837	23	135.057	143.518	2	12	3	
6	7	85986679	86086421	99742	2689	16	97.908	100.961	3	3	6	
7	11	41610583	41712671	102088	3001	90	72.884	61.595	7	15	10	
8	11	113656849	113767585	110736	2941	24	55.851	54.593	3	8	8	
9	17	33966224	34061629	95405	1910	6	55.825	53.234	1	3	2	

*We only considered archaic introgression segments whose length is greater than 100kbp. Regions shown here appear more than 1000 times in results of an overlapping analysis in which all pairs of haplotypes between EUR and SAS are recorded if the overlapping region is longer than 90% of each of the pair. The exact boundaries of each region are identified by the intersection of those records. The numbers of SNVs are according to the original KGP datasets. The TagSNVs should satisfy that alternative alleles don't appear in YRI, but do in Denisova, EUR and SAS.

Figure R3 PCA of the Region chr11:41610583-41712671. We made a PCA of region chr11:41610583-41712671. The left figure includes all samples of the 1000 Genome Project and the right plot only includes European and South Asian samples. The bottom right corner of the two plots display the Denisovan-like sequences, which were likely derived from the same ancestry.

In our manuscript, we also developed a novel statistics Archaic ancestry-sharing (AAS) to explore the relationship of introgressed sequences between two populations (Results). “The ancestry-sharing ratio of Denisovan-like sequences across European populations was much higher than that between any non-European populations in Eurasia, indicating that the Denisovan-like ancestry in the European populations was derived from some indirect gene flow due to recent admixture with other populations. The ancestral source populations were most likely some South Asian groups, as we observed similar levels of ancestry-sharing between European and South Asian populations (on average 51.16, with a minimum of 40.97 and a maximum of 65.40) compared to ancestry-sharing of South Asian populations (on average 48.59, with a minimum of 44.58 and maximum of 61.10). For comparison, the ancestry-sharing ratio in European populations was much higher, with 146.48 on average (a minimum of 122.34 and a maximum of 180.16)”.

Following the reviewer's suggestion, we checked the surrounding area of the Denisovan introgressed fragments and calculated the allele sharing between introgressed fragments and comparable Asian chromosomal segments, we did observe considerable higher allele sharing of the Denisovan introgressed fragments with flanking sequences of Asian ancestry compared with that of European ancestry (**Figure R4**). Therefore, we believe that we have obtained strong evidence that they are of Asian ancestry.

Figure R4 Allele sharing analysis of flanking region of European archaic segments. We compared allele sharing between flanking regions (100 kb each up- and down-stream) of the archaic segments in European and the comparable regions in Asian groups (indicated by light-red color). As the control analyses, we also calculated allele sharing of modern human sequences in comparable regions within Asian groups, within European, and that between European and Asian groups. we did observe considerable higher allele sharing of the Denisovan introgressed fragments with flanking sequences of Asian ancestry compared with

that of European ancestry. The plots numbered 1-9 are corresponding to the Denisovan-like introgressed regions as shown in Table R1.

Comment 3: The identification of old introgressions and their interpretation is interesting. However, I have some concerns about the validity of the data that the authors have used to reach these conclusions. First, the authors exclude Vindija because it uses a different calling method that provides a lower number of SNVs (however, it is used in the example of the Newick notation $((YRI:100,Test:100):557.5,(Denisovan:340,(Vindija33.19:82.5,Altai:20):280):237.5))$ (?). What would happen if the authors had used this genome?

Authors' Response: We thank the reviewer for asking this. Here we confirm that Vindija was not included in our empirical data analysis. The Newick model mentioned by the reviewer was simply presented in the supplementary data as an example to illustrate the Newick format. We apologize for the confusion it may have caused. We have made revisions on S1 & S2, and removed all stuff related to Vindija from SI data.

There was a good reason that we didn't use the Vindija genome in our empirical data analysis. During the high coverage Vindija genome sequencing, the uracil-DNA-glycosylase (UDG) treatment was omitted in the ancient DNA library preparation to obtain more reads. To resolve this problem, Prüfer et al. (Science 358(6363):655-658 2017) developed a new variant calling method snpAD and applied a more stringent variant filtering step (mappability filtration and so on), which leads to a lower number of SNVs identified in that genome. Our analysis showed that the original version of the Altai Neanderthal and Denisovan genomes are relatively less biased and retained many more variants compared with the new version of these archaic genomes (Table R2). Therefore, we used only Altai Neanderthal and Altai Denisovan in the empirical data analysis.

Table R2 Number of SNVs of the Different Versions of Archaic Genomes

Version	Archaic Hominin	0/0	0/1	1/1	1/2	non missing	non 0/0
Kay Prüfer 2014, Nature	Altai Neanderthal	2,629,547,064	1,983,462	4,072,560	6,385	2,635,609,471	6,062,407

Kay Prüfer 2017, Science	Altai Neanderthal	1,762,259,184	384,644	2,387,206	2,462	1,765,033,496	2,774,312
Matthias Meyer, 2010, Science	Altai Denisovan	2,611,580,063	1,808,894	4,169,135	4,778	2,617,562,870	5,982,807
Kay Prüfer 2017, Science	Altai Denisovan	1,761,514,441	417,315	2,474,442	2,234	1,764,408,432	2,893,991
Kay Prüfer 2017, Science	Vindija33.19	1,761,763,557	359,470	2,446,360	1,581	1,764,570,968	2,807,411

However, to investigate whether using Vindija would result in significantly different estimations, we further analyzed both Vindija33.19 Neanderthal and Altai Neanderthal. (Table R3) We did not observe a significant difference in real data analysis. Therefore, using or not using Vindija would not affect our findings as well as conclusions.

Table R3. Archaic Introgression Proportion of Different Continental/regional Populations with Different Matching Models

Dataset	Continent/region	((YRI,Test),(Denisovan,Altai))		((YRI,Test),(Denisovan,Vindija33.19))		((YRI,Test),(Denisovan,(Vindija33.19,Altai)))		
		Denisovan-like (%)	Neanderthal-like (%)	Denisovan-like (%)	Neanderthal-like (%)	Denisovan-like (%)	Vindija-like (%)	Altai-like (%)
KGP	Europe	0.06 (0.04–0.11)	1.09 (0.95–1.26)	0.04 (0.02–0.08)	1.21 (1.05–1.37)	0.03 (0.02–0.05)	1.04 (0.89–1.24)	0.14 (0.10–0.20)
KGP	South Asia	0.12 (0.08–0.17)	1.16 (1.01–1.34)	0.10 (0.05–0.16)	1.27 (1.10–1.48)	0.07 (0.04–0.12)	1.05 (0.91–1.20)	0.15 (0.12–0.20)
KGP	East Asia	0.12 (0.09–0.17)	1.38 (1.24–1.55)	0.10 (0.07–0.15)	1.52 (1.36–1.70)	0.08 (0.04–0.11)	1.29 (1.13–1.44)	0.18 (0.14–0.23)

SGDP	Africa	0.00 (0.00– 0.04)	0.02 (0.00– 0.29)	0.00 (0.00– 0.03)	0.02 (0.00– 0.33)	0.00 (0.00– 0.03)	0.02 (0.00– 0.41)	0.00 (0.00– 0.04)
SGDP	America	0.11 (0.08– 0.12)	1.34 (1.23– 1.45)	0.09 (0.07– 0.10)	1.38 (1.25– 1.49)	0.07 (0.05– 0.09)	1.325 (1.11– 1.33)	0.18 (0.16– 0.21)
SGDP	Central Asia Siberia	0.12 (0.08– 0.15)	1.39 (1.18– 1.52)	0.09 (0.07– 0.12)	1.45 (1.22– 1.61)	0.07 (0.05– 0.09)	1.25 (1.07– 1.39)	0.16 (0.13– 0.21)
SGDP	East Asia	0.13 (0.09– 0.16)	1.44 (1.24– 1.58)	0.10 (0.06– 0.14)	1.49 (1.27– 1.65)	0.08 (0.05– 0.10)	1.29 (1.10– 1.41)	0.16 (0.12– 0.23)
SGDP	Oceania	0.74 (0.12– 0.80)	1.58 (1.18– 1.70)	0.61 (0.09– 0.70)	1.58 (1.24– 1.73)	0.45 (0.07– 0.51)	1.46 (1.09– 1.56)	0.20 (0.14– 0.23)
SGDP	South Asia	0.11 (0.06– 0.16)	1.17 (0.97– 1.36)	0.09 (0.03– 0.12)	1.22 (1.02– 1.39)	0.06 (0.02– 0.10)	1.03 (0.85– 1.16)	0.14 (0.11– 0.19)
SGDP	West Eurasia	0.06 (0.04– 0.09)	1.09 (0.87– 1.36)	0.04 (0.02– 0.07)	1.14 (0.92– 1.39)	0.03 (0.02– 0.05)	0.99 (0.77– 1.17)	0.13 (0.10– 0.17)
EGDP	Africa	0.00 (0.00– 0.00)	0.01 (0.01– 0.02)	0.00 (0.00– 0.00)	0.00 (0.00– 0.00)	0.00 (0.00– 0.00)	0.03 (0.03– 0.03)	0.00 (0.00– 0.00)
EGDP	America	0.09 (0.08– 0.12)	1.27 (1.17– 1.39)	0.08 (0.07– 0.10)	1.36 (1.27– 1.41)	0.06 (0.05– 0.08)	1.22 (1.15– 1.28)	0.18 (0.14– 0.20)
EGDP	Central Asia/Caucasus/Sibe ria	0.11 (0.04– 0.16)	1.33 (0.91– 1.53)	0.09 (0.03– 0.15)	1.43 (0.96– 1.64)	0.07 (0.02– 0.11)	1.21 (0.84– 1.40)	0.16 (0.10– 0.22)
EGDP	North Philippine Negrito (Aeta, Agta, and Batak)	0.53 (0.39– 0.64)	1.34 (1.27– 1.44)	0.58 (0.40– 0.70)	1.54 (1.46– 1.65)	0.42 (0.29– 0.50)	1.36 (1.28– 1.48)	0.19 (0.16– 0.22)

EGDP	South East Asia	0.12 (0.10- 0.13)	1.36 (1.26- 1.44)	0.09 (0.08- 0.11)	1.42 (1.31- 1.50)	0.07 (0.06- 0.08)	1.23 (1.11- 1.30)	0.16 (0.15- 0.19)
EGDP	South Asia	0.09 (0.07- 0.13)	1.05 (0.90- 1.21)	0.09 (0.06- 0.13)	1.21 (1.04- 1.35)	0.07 (0.04- 0.11)	1.05 (0.94- 1.20)	0.15 (0.11- 0.18)
EGDP	West Eurasia	0.06 (0.03- 0.11)	1.14 (0.78- 1.38)	0.05 (0.03- 0.08)	1.22 (0.87- 1.43)	0.03 (0.02- 0.06)	1.04 (0.75- 1.22)	0.14 (0.09- 0.20)

We analyzed data of different continents / regions with different archaic references. In the first series analysis, we used Altai Neanderthal; the second series analysis, we used Vindija33.19 Neanderthal and the third, we used both of the two Neandertal genomes.

Comment 4: Is the method dependent on the SNV calling (I guess so, even if it is stated in S2.7 that it is robust to errors)? In that case, which kind of SNV calling pipeline is applied to ensure that the different datasets are called in the same way? Similarly, how is the deep coverage at each position taken into account when calling each SNV?

Authors' Response: We thank the reviewer for this comment. We share the concern of the reviewer as we were aware that there could be a potential effect of the SNV calling pipelines because we used three data sets of different sources. Consequently, we have been taking much caution to ensure the data quality in our analysis. First, we checked the details of data processing of different data sources. The EGDP data were quality controlled by using only the concordant variant between genotyping array and sequencing data, sufficient sequencing coverage and T_i / T_v ratio were also applied to ensure data quality. The KGP data processing was also stringently quality-controlled using various parameters such as $AC \geq 2$ and SVM score of ≥ 0.78 .

To further evaluate potential batch effects between different datasets, we examined the potential batch effects in the data by PCA and ADMIXTURE analysis and it turned out such a batch effect was little (see Figures R5 and R6). In the PC plots, different colors stand for different continents and different shapes stand for different datasets. We observed that data were clearly clustered by colors, not by shapes. The

introgression proportions are largely consistent between populations in the same continent (Table S5.1-5.3).

Figure R5 PCA of the Four Datasets. We applied PCA to the four datasets with different continents populations: **a** global population; **b** African populations; **c** American populations; **d** South Asian populations; **e** East Asian populations. Here different shapes stand for different datasets: circle stands for EGDP; rectangle for KGP, diamond for SGDP, and triangle for Ust.

Figure R6 ADMIXTURE Analysis of the Four Datasets. Here each individual is represented by a vertical line, which was partitioned into segments corresponding to the ancestral clusters indicated by color. Samples were grouped according to their population labels. Colors of the x-axis indicate different datasets.

In addition, to mitigate the impact of sequencing errors, we used the intersection SNPs set among Archaic reference, Africans and test non-Africans in our method. Since most of the sequencing errors are likely to be population specific (i.e., singletons or low frequency variants), which were filtered out in our analysis thus were not be presented in the introgressed sequences detection, our method is less likely to be affected by potential sequencing errors in the data. Moreover, in our method, the most informative variants are State 2 (archaic markers, Figure S2.1), i.e., they are the mutations which could be found in archaic reference and the tested non-Africans but not in Africans. They are less likely affected by the calling pipeline. Even though some error sequenced / called variants were introduced in our analysis, HMM and “Segments Connection and Filtration” step will be aware and make a relatively accurate inference. HMM is a model to find similar sequences with a certain tolerance of differences and the “Segments Connection and

Filtration” step is used to connect adjacent archaic sequences, which are separated by erroneous inferred modern sequence, and filter out shorter archaic sequences which are more likely affected by sequencing errors or mapping errors. We believe our method is robust to sequencing errors especially considering the error rate $< 0.1\%$ in NGS data, because sequencing errors are expected to be randomly and sparsely located in the genome, which do not affect FPR and have little influence on the power of matching algorithm.

Comment 5: This extends to the application of SHAPEIT2 to Ust’Ishm and the SGDP. It is known that SHAPEIT2 can produce artifacts when the reference populations (1000K) differ from the target populations (for example, Papuans or an ancient DNA sample). How were the noncalled SNVs treated? I really hope they were not imputed by SHAPEIT2. Also, how was the derived state of each SNV estimated?

Authors’ Response: We thank the reviewer for this comment. We confirmed here that we did not include imputed variants by SHAPEIT2, as we removed the missing genotypes in Ust’Ishm and the variants of lower call rate in the SGDP populations. For the non-called variants in any dataset, we simply removed them as we only considered the interested variants. We determined the derived state of each SNV according to an ancestral state file (-A option to specify). In our empirical data analysis, we used the 71 release of the Ensembl ancestral alleles (http://ftp.ensembl.org/pub/release-71/fasta/ancestral_alleles/) to determine it.

Comment 6: Furthermore, in supplementary materials it is stated “conducted a joint calling with another 1508 unpublished sequencing data generated by our group”. Is this dataset used in this study? If not, which is the purpose of using it for the calling?

Authors’ Response: We thank the reviewer for this comment. We confirm here that we did not use the 1508 unpublished sequencing data in this study. The purpose of using the joint calling of this dataset together with SGDP samples is to improve the power and accuracy of the variant calling in the SGDP data.

Comment 7: I think that the authors should first prove that the joint dataset that they use is not biased in one way or another. I would suggest checking that samples from the same geographic region (but from different datasets) cluster together independently of dataset in classical PCA/ADMIXTURE analyses, as well as that the results from their software are not dataset dependent (for example, for classical PCA/ADMIXTURE analyses, as well as that the results from their software are not dataset dependent (for example, for each geographic region check that the data from the different datasets provide similar results of introgression(not clear checking S5.1...)).

Authors' Response: Following the suggestion of the reviewer, we evaluated the potential batch effect in the dataset with PCA and ADMIXTURE and concluded that there is no obvious batch effect between different datasets (Fig R3-4). As indicated in our response to Comment #4 above, the introgression proportions are consistent between populations in the same continent (Table S5.1-5.3).

Comment 8: In addition, have the authors assessed the power of the method against simulated datasets that include some of the scenarios for which the method should be robust (modern contamination, sequencing errors, etc)?

Authors' Response: We thank the reviewer for this comment. We did not include these scenarios in our simulation study. Our method has no power to detect the introgressed sequences in case modern contamination happened in both Denisovan and Neanderthal references at the same genomic position. However, we believe that the likelihood of this scenario is very small. We also believe our method is robust to sequencing errors especially considering the error rate $< 0.1\%$ in NGS data, because sequencing errors are expected to be randomly and sparsely located in the genome, which do not affect FPR and have little influence on the power of matching algorithm.

Another main point that I think the authors should clarify is the benchmarking:

Comment 9: 1) The simulated datasets are constructed with ms. This software is not providing the introgressed regions and the authors implemented a script to infer the topology of the tree (SimAncestry) at a given genomic region (I guess, no information is provided about how this algorithm works). How is this script dealing with incomplete

lineage sorting? How dependent is this inference to the SNV density in the genomic region? In other words: to which extent are the authors not biasing their conclusions by imputing the ancestry of these regions by means of SimAncestry? Or, alternatively, why not directly applying SimAncestry to the real data?

In any case, I think it is important to provide a much more detailed explanation of this software, given that the benchmarking is basically done with its output.

Authors' Response: We thank the reviewer for this comment. We clarify here that SimAncestry is only a simple script to analyze the tree structure outputted by ms. It cannot be applied to the real data as it was not designed to work on SNP data. The algorithm used in SimAncestry is as follows. For each non-African node, it makes a transversal up to the root and checks if this node coalesces to archaic lineage before to any African lineages. If yes, it determines the node to be introgressed. In this study, this script was used only to obtain the ground truth for introgressed sequences.

Comment 10: 2) The authors must provide the references for all the parameters that are used in the simulations. The time of the Altai Neanderthal seems to be much older than the one reported in the original publication (Prufer et al, ~50kya in contrast to the 60kya of the simulations and the same for Denisovans (in contrast to the 80kya)).

Authors' Response: We thank the reviewer for this comment. For the Altai Neanderthal, the bone age is 50,300 +/- 2,200 years (abstract of Prufer et al. Nature 2014) and the estimated branch shorten is 64 ~ 68 kya (table 1 of Prufer et al. Nature 2014). Considering these, we used a close value of 60 kya. For the Denisovans, Meyer et al. estimated the branch shorted is 74 ~ 92 kya (Figure 2 of Meyer et al. Science 2012) and Prufer et al [Science 2017] got a similar result 70 ~ 90 kya. Hence, we took a median value of 80 kya.

Comment 11: 3) Related to the previous point, can the authors show that the genetic variation obtained in the simulations resemble the genetic variation observed in real populations? I am asking this question because the effective population sizes that the authors are providing do not match any of the described values in the literature (i.e. 1M chromosomes for Non African). I understand that these values are obtained after a phase

of exponential growth, but in any case it is a substantial amount of chromosomes and I guess that this can influence the power for detecting introgressed fragments.

Authors' Response : We thank the reviewer for this comment. The simulation parameters we used were adapted from a study by Vernot and Akey (Science 343(6174):1017-1021). Further description of simulation parameters was provided in the SI of their study (Page 2-3). In these simulation scenarios, the final (present-day) effective population size is 512,000 for European and 1,370,990 for East Asian.

We also did some additional simulations with smaller N_e (7000 for non-African modern humans, Skov et al. 2020). The mean value of Precision is 94.72% (92.47% ~ 96.06%), TPR is 84.37% (80.45% ~ 88.01%) and FPR is 0.50% (0.32% ~ 0.76%) (Figure R1). Theoretically speaking, increasing the effective population size of non-African modern humans does not increase the power of our method. The most informative variants are the state 2 variants, which are absent from African, but shared among introgressed sequences and archaic hominins. The number of this kind of variant will not increase when the modern human effective population size becomes larger, and only the divergence time can influence that number. This is further confirmed by further simulation results (Figure R1).

Comment 12: 4) A bigger concern is the provided ms code. I understand (from the ms code) that the authors are simulating 10Mb each time. However, this information is not present (at least I have not been able to find it) in the description of the simulations.

Following the ms code, I am not sure I understand the effective population size of Denisovans (3 and 4) and Neanderthals (5 and 6) (among other pops):

-n 3 1e-10 -n 4 1e-10 -n 5 1e-10 -n 6 1e-10 -n 7 1e-10 -n 8 1e-10

Similarly, I cannot find in the ms code that Denisovan and Altai Neanderthal are ancient samples (-eA option of ms)(?). Does this mean that they are sampled from the present in the simulations?

Another problem is the growth rate: -eg 1e-10 1 175.328. In ms time is specified in terms of $4 \cdot N_e$ generations. What does a scaled time of $1e-10$ mean in the simulations (according to the SI, the time of the second expansion is TAMHexp2=5kya and of the first TAMHexp1=23kya)? Similarly, a growth rate of 175.328 is extremely big. All this will

increase the number of rare mutations, enhancing the possibility to distinguish between populations.

Authors' Response: We thank the reviewer for this comment. We would like to clarify that the related information was present in the supplementary (i.e., the 3rd paragraph of S3): “We simulated 10 Mb genomes for each run. The mutation rate was set to 1.25×10^{-8} per generation per bp. The recombination rate was set to 1×10^{-8} .” As far as we are aware, there is no “-eA” option in the software ms. To circumvent this problem, we use an approach suggested by Fu et al. (*Nature* 514:7523, 445-449, 2014). That is, we set the N_e for pop 3-8 as $1e-10$ (an extremely small value) at time 0 (present). In the simulation, this extremely small N_e was changed to the archaic N_e when the archaic sample died. This is to ensure these lineages have an extremely small mutation and recombination during this period, just like they are dead. We also did some additional simulations with smaller N_e (7000 for non-African modern humans, Skov et al. 2020). The mean value of Precision is 94.72% (92.47% ~ 96.06%), TPR is 84.37% (80.45% ~ 88.01%) and FPR is 0.50% (0.32% ~ 0.76%) (Figure R1).

Comment 13: 5) Generation time. This is not a parameter that one applies to ms and it should be specifically stated in the main text, in addition to the mutation rate and recombination rate (with a proper reference in all cases).

Furthermore, despite there is no clear standard for the generation time, it is usually in the range of 29-30 years/generation using Hunter-Gatherers (i.e. Fenner 2005). This could be a potential problem for the estimates that the authors provide, based on a 25 years per generation (which has also been claimed in other studies but at least the authors should cite them). Using 29 years/generation, a much older time estimates are obtained. However, this could be partially explained if other (higher) mutation rates were used.

Authors' Response: We thank the reviewer for this comment. As there is no standard for the generation time, we always provide time estimation in both generations and years. Following the suggestion, we have added a reference about the generation time we used (i.e., Fenner 2005) in the main text. Since our simulation results are not affected by the

generation time, we used 25 years per generation. However, in real data analysis, we applied 30 years/generation so that the results could be comparable to the literature.

We also did some additional simulations with 30 year / generation. The results show that this change does not have any noticeable influence on the performance of our method. The mean value of Precision is 94.72% (92.47% ~ 96.06%), TPR is 84.37% (80.45% ~ 88.01%) and FPR is 0.50% (0.32% ~ 0.76%) (Figure R1).

Comment 14: Another point is that it is known that Denisovans and Neanderthals admixed, both between and also with other archaic populations. How would this affect the inference of introgression and time?

Authors' Response: We thank the reviewer for this comment. As the reviewer might be also aware, some parts of archaic genomes were derived from other archaic hominins or even modern humans (a kind of indirect introgression via recent gene flow between present-day populations). This impure archaic reference will lead to the underestimation of introgressed sequences. It is challenging to identify the introgressed sequences in the admixed genomic regions. However, assuming these genomic replacements in the archaic genomes to distribute randomly, the underestimation of the introgression in the genome would be also random. It is thus expected to have little influence on the segment length distribution and time estimation. Also, it is expected to not or less likely influence the total amount of the introgressed sequences.

Other minor points:

Comment 15: 1) What does "The introgression time and introgression proportion of each wave changed in gradient" mean?

Authors' Response: We thank the reviewer for asking for this. Here, we were trying to describe how different parameters for introgression time and proportion were set for each wave: both of them are changed in a grid-like fashion. We have rephrased this sentence in the revised version.

Comment 16: 2) I am not convinced about this sentence "Simulation results showed that ArchaicSeeker2.0 have a high precision (~93.0%), a high true positive rate (~90.4%) and

a low false positive rate (~0.14%) in the archaic segments detection. In most simulations, more than 80% introgressed segments matched to the correct ancestry. For the introgression history inference, ArchaicSeeker2.0 inferred the correct scenarios for most cases (122/144, 84.7%).” How likely is that, if the authors distribute at random the fragments in the simulated 10Mb, they observe the same amount of overlap as with the inferred fragments?

Authors’ Response: We thank the reviewer for this comment. Following the suggestion, we distribute at random the fragments in the simulated data by splitting them into 10kb bins and random shuffling. Then we applied our method to these random data. We repeated these processes 100 times. We observed a clear drop of TPR and precision and obvious increase of FPR in the random shuffling data, i.e., TPR is only 48.5% (35.4 ~51.5%), precision is 81.1% (78.4 ~ 88.7%), and FPR is 0.236% (0.093 ~ 0.285%), compared with previous TPR 90.8%, precision 93.7%, and FPR 0.124%. The reason was that the length distribution of these introgressed sequences was significantly changed in the random shuffling data. These results also indicated that the fragment length distribution is very important to the power and accuracy of our method.

Comment 17: 3) A related question is if the percentage of inferred introgression in the genome correlates with the amount of introgression (in principle, the correlation should be quite high if there are no problems with the simulations).

Authors’ Response: We thank the reviewer for this comment. We can confirm that the correlation is rather high, as shown in Table S4.13 and S4.14. For example, the introgression proportion simulation parameter α is almost identical to the inferred α .

Comment 18: 4) S4.1 and S5.1.. What happens if the software is run with different hyperparameters (i.e. instead of considering introgression at 2% we run at 10%)?

Authors’ Response: Many thanks for this interesting question. Since our software could automatically estimate the parameters, we tested the introgression parameter with simulation data. For scenarios with different introgression proportions, ranging from 0.2% ~ 4%, we only use 2% (the default parameter) as the initial value and our method

converges to the correct value fast. For instance, when the alpha is set as 10%, the parameters of the software converge to the same parameter set in just two iterations (Table R4).

Table R4 HMM Parameter Estimation with Abnormal Initial Values

	Initial Value	iteration 1	iteration 2	iteration 3
Alpha	0.02	0.0178992	0.0180233	0.0180233
T	2000	1621.3	1604.69	1604.69
Alpha	0.1	0.0177725	0.0180233	0.0180233
T	2000	1655.56	1604.69	1604.69

Comment 19: 5) S4.1.2. Unclear what extra information is providing the usage of SNPs since these are ultimately used for defining the introgressed fragments. For the Non-African AIMS, the statistics of the performance are computed considering only the Non-African AIMS or assuming all the SNPs?

Authors' Response: We thank the reviewer for asking this. Yes, at this step, we only evaluate the accuracy of the Non-African AIMS, which are also the most informative information in our method.

Comment 20: 6) “The total length test non-African genome of 90 repeats is $90 * 200 * 10 = 1.8 * 10^{11} \text{bp}$ ” in SI This sentence is wrong. I guess the authors mean $90 * 200 * 10^7$

Authors' Response: We thank the reviewer for pointing out this typo. Indeed, there should be a ‘Mb’ after 10, i.e., 10 Mb or 10^6 bp . We have corrected this in the revised version.

Comment 21: 7) It looks to me that many of the times of introgression are systematically underestimated or overestimated (STable 4.15).

Authors' Response: We thank the reviewer for this comment. We agree that there is some underestimation or overestimation of these times. However, we think most of them

should be reasonably close to the ground truth value when the number of introgression events is correctly inferred.

Comment 22: 8) It is unclear why an EM method is more “friendly and robust” to estimate the parameters. In my opinion, the EM algorithm should be quantified in terms of convergence towards the true value.

Authors’ Response: We thank the reviewer for this comment. Here, we would like to emphasize that our method can estimate the model parameters automatically, which is our motivation to use “friendly and robust”. Here we set a convergent criterion, as described in the last paragraph of S2.5. *“After the updating of the parameters, we repeated the seeking algorithm and the connection procedure. Until the absolute value of difference between the updated T and previous T less than 0.01, we finished the parameter estimation iteration.”*

”

Comment 23: 9) The paper claims that it is not model dependent, but it is obvious that it requires a basic demographic model to be fitted (nonadmixed outgroup, relationship of the archaic populations).

Authors’ Response: We thank the reviewer for this comment. However, we are not aware of such a description in our manuscript and we have checked our manuscript again to make sure that this is not contained in the text. We did mention that our method is “flexible and powerful” and “more user friendly and more robust”. Indeed, our method required a rough phylogenetic relationship / topology among analyzed populations, but we calibrated it based on the input data before we performed the matching algorithm. Furthermore, the method does not use any information from this model to infer the introgression history.

Comment 24: 10) The section of the Matching Model is confusing. For what I see, it is not taking the effective population size of the different populations into account to estimate the time of divergence.

Authors' Response: We thank the reviewer for this comment. Here we would like to confirm that the effective population size is considered in our method (Equation 2.10): where the parameter (N_e) could be regarded as a part of the population level mutation rate ($\theta = 4N_e\mu$) and our method estimated this parameter automatically.

Comment 25: 11) The SI and Main text will require extensive English editing.

Authors' Response: We have made substantial editing on both the SI and main text with help of native English speakers by using a commercial service (Letpub) for language editing.

Authors' Response to Reviewer # 2 (Remarks to the Author):

The authors introduce a new method for detecting archaic introgression and dating the admixture times. The authors have performed a lot of tests and have done a good job of documenting them. But I fear its the wrong kind of tests and I am not convinced this method works as well as the authors claim it does.

Major concerns

Comment 1: My main concern is that many of the admixture proportions and admixture times do not fit with literature.

For instance they detect a second wave of Neandertal admixture into European populations which occurred between 37.5-33.0 thousand years ago. All the evidence I have seen dates the last known Neandertals around 40 thousand years ago (Bard et al 2020, Higham et al 2014). The date of admixture is also substantially younger than what is estimated elsewhere (Moorjani 2016, Sankararaman 2016).

If the authors are correct this would be a groundbreaking claim - the authors need to comment on this!

Authors' Response: We thank the reviewer for this comment. We agree that our results indicate that the dates we inferred are younger than those suggested in several previous studies. Sankararaman S, et al, (PLoS Genet.8(10):e1002947, 2012) used the

extent of linkage disequilibrium (LD) in the genomes of present-day Europeans and found that the last gene flow from Neandertals (or their relatives) into Europeans likely occurred 37,000–86,000 years before the present (BP). Our estimations (37.5-33.0 thousand years ago) are slightly lower but still close to this relatively wide range of estimation.

In addition, there are also some previous studies suggesting more recent Neanderthals in Europe. For example, Zafarraya Neanderthal mandible was found in a cave (Cueva del Boquete) in 1983 by Cecilio Barroso and Paqui Medina. The mandible has been dated to 30,000 years Before Present (BP), and at the time represented the youngest-known Neanderthal remains (Hublin J.J., C. R. Acad. Sc. Paris). In Gorham's Cave, researchers even found the survival of a population of Neanderthals to 28 kyr BP (Finlayson et. al., Nature 443:850–853, 2006).

Comment 2: Another instance is the amount of identified Denisovan sequence in Papuans is very low! The authors estimate around 0.41%– 0.73%. From Dstatistics (3-6% Prüfer et al 2014), other methods (>2% Denisovan Skov et al 2018). This suggests to me that the method is highly affected by how close the introgression denisovan is to the sequenced Denisovan. And if it's the case that the authors only pick up a fraction of all the Denisova sequence which is there (the part that is most similar to the sequenced Denisovan genome presumably), any claim about admixture timing becomes difficult.

Authors' Response: We thank the reviewer for this comment. We agree that the Denisovan-like introgression ancestry proportion in Papuan detected in this study is much lower compared with the results reported by several previous studies. However, in our simulation, which is based on simulations in a study by Vernot and Akey (Science 343(6174):1017-1021, 2014), there is no such significant underestimation of the total archaic ancestry (Table S4.13, S4.14).

Considering the reviewer's concerns about our benchmarking (see Comment 3 below), we performed a series of simulations based on the model specified in Skov et al (Nature, 582:78-73, 2020), that is, assuming 8% Denisovan-like introgression and 2% Neanderthal introgression. We simulated 10 Mb sequences of 100 Africans (200 haplotypes), 100 Non-Africans (200 haplotypes), 1 Neanderthal (2 haplotypes) and 1 Denisovan (2 haplotypes) and repeated it 100 times. We inferred totally ~8.54% archaic

introgressed sequences. We further evaluated the accuracy of our results by comparing the correctly inferred lengths. The mean value of Precision is 94.72% (92.47% ~ 96.06%), TPR is 84.37% (80.45% ~ 88.01%) and FPR is 0.50% (0.32% ~ 0.76%) (Figure R1). The number interval in the brackets is the 95% CI. As you can see, there is some slight underestimation of the total archaic ancestry compared with that was set in the simulation (10%). Therefore, there is some slight underestimation of the archaic sequences by our method in these simulations, but definitely not that low.

To figure out the root of the inconsistency between our estimation and that of some previous studies, we made a systematic investigation of the literature and studied the details of the methodology as well as the data type used. It turned out that most of the site-based methods which were originally designed for analyzing genotyping data, such as D-Statistics, F-Statistics make a higher introgression estimation, For example, David Reich et al. (Nature 468, 1053–1060, 2010) used D-Statistics and estimated Denisovan 4–6% of its genetic material to the genomes of present-day Melanesians. Benjamin Vernot et al. used the f_4 statistic and found significant evidence of Denisovan ancestry ($Z > 4$) in Melanesian samples, with admixture proportions varying between 1.9 and 3.4%.

On the contrary, sequence-based methods detected much less archaic ancestry compared with that estimated by the site-based methods. For instance, Skov et al (PLOS Genetics 14(9) Table 1) presented a summation of the lengths of Denisovan-like introgressed sequences from different methods. Taking Papuan as an example, HMM-based method detected 83.11 Mb, Sstar detected 43.11 Mb, CRF detected 58.17 Mb, and Sprime detected 38.98 Mb Denisovan sequences. These results are comparable to ours, i.e., 43.8 Mb. The “seemingly” underestimation might due to the differences between different types of methods. Some degree of the difference also attributes to data types, i.e., early studies mainly analyzed genotyping data generated by SNP array, more recent studies had more chances to analyze deep-sequencing genomes. For example, Sankararaman et. al. [Table 1 in Curr Biol. 2016 May 9;26(9):1241-7] reported just $0.85 \pm 0.43\%$ Denisovan introgression in Oceanian using the f_4 test.

In our simulation study, we do simulate scenarios of deep divergent lineages / unknown archaic introgression (S4.3). The history analysis is quite consistent with the ground-truth introgressed time (The last four rows of Table S4.13). In other words, the

length of mismatched Denisovan (some sequences seem not there or unlike the sequenced Denisovan) introgressed sequences should be random. Therefore, excluding the mismatched sequences does not change the distribution of introgressed sequence lengths and will not affect our following admixture timing analysis. Regarding the introgression time estimation, we did a series simulation to demonstrate that this kind of slight underestimation has little influence on the results (S4.2).

Comment 3: I have looked at the simulation studies, because the results from this is the main argument that the method works well. Here I have some major concerns. The simulations are in a sense too well behaved. The population sizes are small (like the ancestral population size of modern humans is 3000 - but should probably be more like 7000 Gutenkunst et al 2009 - so things coalesce quickly), the recombination map is constant and it's assumed that one can call genotypes perfectly at every position in the Neandertal and Denisova genomes. In order to really assess their model the authors absolutely need to do the following things:

Comment 3.0: 0) Use msprime instead of ms commands - it's much easier to use and maintain the code for.

Authors' Response: We thank the reviewer for this suggestion. Although msprime is much easier to use, we are more familiar with ms and converting the output of msprime to our data analysis pipeline remains a challenge. Given these practical considerations and the fact that simulation is time-consuming, we would like to ask your permission to continue to use ms for this study.

Comment 3.1: 1) increase effective population sizes (I would recommend the demography from Skov et al 2020 which is based on the Gutenkunst paper)

Authors' Response: We thank the reviewer for this comment. We further performed a series of simulations based on the model specified in Skov et al 2020, assuming 8% Denisovan-like introgression and 2% Neanderthal introgression. We simulated 10 Mb sequences of 100 Africans (200 haplotypes), 100 Non-Africans (200 haplotypes), 1 Neanderthal (2 haplotypes) and 1 Denisovan (2 haplotypes) and repeated it 100 times. We totally inferred ~8.54% archaic introgressed sequences. We evaluated the

performance of our method based on the correctly inferred segment length. The mean value of Precision is 94.72% (92.47% ~ 96.06%), TPR is 84.37% (80.45% ~ 88.01%) and FPR is 0.50% (0.32% ~ 0.76%) (Figure R7). The number interval in the bracket is the 95% CI. As you can see, there is some slight underestimation of the total archaic ancestry in the simulation (10%). These results suggested that although there might be some slight underestimation in our method, the main results derived from our method should be reasonable.

Figure R7 Length based evaluation with Skov's simulation scenarios. Comparison between the inferred introgressed segments and the ground truth segments under the scenarios described in Skov et al 2020. The y-axis represents the summary statistics of precision, TPR and FPR.

Comment 3.2: 2) add varying recombination rate!

Authors' Response: Following the suggestion, we additionally performed a series of simulations with different recombination rates. Since the average recombination rate is around 1.2 cM/Mb, we set the recombination rate from 10%-90% with a step of 10% to the average recombination rate. We also performed two series scenarios with higher recombination rates: 2X and 4X. As shown in Figure R8, our method has a better performance when the recombination rate is low. When the recombination rate is just 10% of the average values, the precision is ~98.6%, TPR is 91.8% and FPR is 0.11%. However, the precision drops to 85.4% and TPR to 66.7%, the FPR rises to 1.26% when the recombination rate is 4X the average recombination rate. These results suggested that our method performs well in real data analysis because recombination is not evenly distributed on the genomes and most of the genomic regions are of low recombination rate.

Figure R8 Length based evaluation with scenarios of different recombination rates. Comparison between the inferred introgressed segments and the ground truth segments under the scenarios with different recombination rates. The y-axis represents the summary statistics of precision, TPR and FPR.

Comment 3.3: 3) mask out snps in the archaic sequences that fall outside the mappable regions (<https://bioinf.eva.mpg.de/map35/100/>)

Authors' Response: We thank the reviewer for this suggestion. We have considered this issue and believe it is better not to apply this mappable region mask to the empirical data for this particular study. Our decision is based on the following two reasons. First, this genome mask will remove some or part of some introgression regions in our data. Indeed, in a preliminary study we did find some introgressed regions which were filtered out or interrupted by the mask regions. This will affect the length distribution of these introgressed sequences and the following history reconstruction. Secondly, our method uses HMM to detect the candidate introgressed sequences, which does not rely on single variants and thus is less likely affected by sequencing or mapping errors.

In addition, such mappable masks were not performed for the first paper of the high coverage sequencing study of Denisovan or Altai Neanderthal [more details about

the paper]. The authors did this kind of mask in the high coverage sequencing study of the Vindija genome. One of the reasons this mask was done is that UDG-treatment was not used on the ancient samples and they implemented a new calling method snpAD. As we did not use the Vindija genome in our analysis, we think such stringent filtration is not required for our analysis.

Comment 3.4: Rerun all the analysis with this new demography and evaluate the method on these simulations instead. Before we know how well the method works on these more realistic simulations it does not make sense to evaluate any claims made about admixture times and admixture proportions.

Authors' Response: We thank the reviewer for this suggestion. We would like to clarify that our simulations follow the same simulation demography and parameters as reported in a paper by Vernot and Akey (Science 343(6174):1017-1021).

Minor comments

Comment 4:# Alien DNA? I would call it ancient hominin/archaic DNA.

Authors' Response: Following the suggestion, we have changed the word in the revised version.

Comment 5: # Ancient DNA

The authors does not include the Vindija genome because the number of SNPs is much less than the 2 previously published genomes - but there is no citation for this? How much less data? The Vindija Neandertal is much closer to the introgressing Neandertal than the Altai Neandertal and since the detection method relies on matching with archaic DNA I would think the segment determination would be more accurate using this genome.

Later the authors write:

In our method, we used African whole genome sequencing data (YRI) and archaic high coverage whole genome sequencing data (Altai Denisovan, Altai Neanderthal or Vindija33.19) as the references.

So are you using the Vindija genome or not?

Authors' Response: We thank the reviewer for asking this. The reason that we didn't use the Vindija genome in our empirical data analysis is as follows. The Vindija high coverage genome omits the uracil-DNA-glycosylase (UDG) treatment in their ancient DNA library preparation to obtain more reads. Instead, they developed a new variant calling method snpAD and applied a more stringent variant filtering step (mappability filtration and so on), which makes the number of SNVs lower in that genome (Prüfer et al, Science 358(6363):655-658, 2017). We believe that the original version of the Altai Neanderthal genome should be more reliable and have more variants compared with the Vindija genome, so we used only Altai Neanderthal in the empirical data analysis. In the Vindija high coverage studies, they used snpAD to redo the calling of Altai Neanderthal and Altai Denisovan. We compared the number of SNVs between the two versions of Altai Neanderthal. More than half of the non-homozygous reference alleles were filtered out in the new version (6,062,407 vs. 2,774,312). For the heterozygous alternative alleles, only around 20% remained in the new version.

Table R5 Number of SNVs of the Different Two Versions of Archaic Genomes

Version	Archaic Hominin	0/0	0/1	1/1	1/2	non missing	non 0/0
Kay Prüfer 2014, Nature	Altai Neanderthal	2,629,547,064	1,983,462	4,072,560	6,385	2,635,609,471	6,062,407
Kay Prüfer 2017, Science	Altai Neanderthal	1,762,259,184	384,644	2,387,206	2,462	1,765,033,496	2,774,312
Matthias Meyer, 2010, Science	Altai Denisovan	2,611,580,063	1,808,894	4,169,135	4,778	2,617,562,870	5,982,807
Kay Prüfer 2017, Science	Altai Denisovan	1,761,514,441	417,315	2,474,442	2,234	1,764,408,432	2,893,991
Kay Prüfer 2017, Science	Vindija33.19	1,761,763,557	359,470	2,446,360	1,581	1,764,570,968	2,807,411

However, to investigate whether using Vindija would result in significantly different estimations, we further analyzed both Vindija33.19 Neanderthal and Altai Neanderthal (Table R2). As we did not observe a significant difference in real data analysis, using or not using Vindija should not affect our findings as well as conclusions. Therefore, we used only Altai Neanderthal in the empirical data analysis to keep more variants.

Table R2. Archaic Introgression Proportion of Different Continental/regional Populations with Different Matching Models

Dataset	Continent/ region	((YRI,Test),(Denisovan,Altai))		((YRI,Test),(Denisovan,Vindija 33.19))		((YRI,Test),(Denisovan,(Vindija33.19,Altai)))		
		Denisovan- like (%)	Neanderthal- like (%)	Denisovan- like (%)	Neanderthal- like (%)	Denisovan- like (%)	Vindija-like (%)	Altai-like (%)
KGP	Europe	0.06 (0.04– 0.11)	1.09 (0.95– 1.26)	0.04 (0.02– 0.08)	1.21 (1.05– 1.37)	0.03 (0.02- 0.05)	1.04 (0.89- 1.24)	0.14 (0.10- 0.20)
KGP	South Asia	0.12 (0.08– 0.17)	1.16 (1.01– 1.34)	0.10 (0.05– 0.16)	1.27 (1.10– 1.48)	0.07 (0.04- 0.12)	1.05 (0.91- 1.20)	0.15 (0.12- 0.20)
KGP	East Asia	0.12 (0.09– 0.17)	1.38 (1.24– 1.55)	0.10 (0.07– 0.15)	1.52 (1.36– 1.70)	0.08 (0.04–0.11)	1.29 (1.13–1.44)	0.18 (0.14- 0.23)
SGDP	Africa	0.00 (0.00– 0.04)	0.02 (0.00– 0.29)	0.00 (0.00– 0.03)	0.02 (0.00– 0.33)	0.00 (0.00–0.03)	0.02 (0.00–0.41)	0.00 (0.00- 0.04)
SGDP	America	0.11 (0.08– 0.12)	1.34 (1.23– 1.45)	0.09 (0.07– 0.10)	1.38 (1.25– 1.49)	0.07 (0.05–0.09)	1.325 (1.11–1.33)	0.18 (0.16- 0.21)
SGDP	Central Asia Siberia	0.12 (0.08– 0.15)	1.39 (1.18– 1.52)	0.09 (0.07– 0.12)	1.45 (1.22– 1.61)	0.07 (0.05–0.09)	1.25 (1.07–1.39)	0.16 (0.13- 0.21)
SGDP	East Asia	0.13 (0.09– 0.16)	1.44 (1.24– 1.58)	0.10 (0.06– 0.14)	1.49 (1.27– 1.65)	0.08 (0.05–0.10)	1.29 (1.10–1.41)	0.16 (0.12- 0.23)
SGDP	Oceania	0.74 (0.12– 0.80)	1.58 (1.18– 1.70)	0.61 (0.09– 0.70)	1.58 (1.24– 1.73)	0.45 (0.07–0.51)	1.46 (1.09–1.56)	0.20 (0.14- 0.23)
SGDP	South Asia	0.11 (0.06– 0.16)	1.17 (0.97– 1.36)	0.09 (0.03– 0.12)	1.22 (1.02– 1.39)	0.06 (0.02–0.10)	1.03 (0.85–1.16)	0.14 (0.11- 0.19)
SGDP	West Eurasia	0.06 (0.04– 0.09)	1.09 (0.87– 1.36)	0.04 (0.02– 0.07)	1.14 (0.92– 1.39)	0.03 (0.02– 0.05)	0.99 (0.77–1.17)	0.13 (0.10- 0.17)

EGDP	Africa	0.00 (0.00-0.00)	0.01 (0.01-0.02)	0.00 (0.00-0.00)	0.00 (0.00-0.00)	0.00 (0.00-0.00)	0.03 (0.03-0.03)	0.00 (0.00-0.00)
EGDP	America	0.09 (0.08-0.12)	1.27 (1.17-1.39)	0.08 (0.07-0.10)	1.36 (1.27-1.41)	0.06 (0.05-0.08)	1.22 (1.15-1.28)	0.18 (0.14-0.20)
EGDP	Central Asia/Caucasus/Siberia	0.11 (0.04-0.16)	1.33 (0.91-1.53)	0.09 (0.03-0.15)	1.43 (0.96-1.64)	0.07 (0.02-0.11)	1.21 (0.84-1.40)	0.16 (0.10-0.22)
EGDP	North Philippine Negrito (Aeta, Agta, and Batak)	0.53 (0.39-0.64)	1.34 (1.27-1.44)	0.58 (0.40-0.70)	1.54 (1.46-1.65)	0.42 (0.29-0.50)	1.36 (1.28-1.48)	0.19 (0.16-0.22)
EGDP	South East Asia	0.12 (0.10-0.13)	1.36 (1.26-1.44)	0.09 (0.08-0.11)	1.42 (1.31-1.50)	0.07 (0.06-0.08)	1.23 (1.11-1.30)	0.16 (0.15-0.19)
EGDP	South Asia	0.09 (0.07-0.13)	1.05 (0.90-1.21)	0.09 (0.06-0.13)	1.21 (1.04-1.35)	0.07 (0.04-0.11)	1.05 (0.94-1.20)	0.15 (0.11-0.18)
EGDP	West Eurasia	0.06 (0.03-0.11)	1.14 (0.78-1.38)	0.05 (0.03-0.08)	1.22 (0.87-1.43)	0.03 (0.02-0.06)	1.04 (0.75-1.22)	0.14 (0.09-0.20)

We analyzed data of different continents / regions with different archaic references. The first series analysis used Altai Neanderthal, the second series analysis used Vindija33.19 Neanderthal, and the third one used both of the two Neandertahls.

Comment 6: # Phasing of data.

For those SGDP populations which are not well represented by the 1000 genomes populations phasing is going to work less well. This will surely effect the how well you can determine archiac sequences?

Authors' Response: We agree with the reviewer that overall phasing does not work for SGDP data as well as for the 1000 genomes. However, we actually found that the current phasing algorithm has a very good performance for the introgressed regions, especially for archaic sequences. The especially good performance mainly attributes to the extreme long divergent time between modern human and archaic hominins, which contributed to highly-divergent alleles between the introgressed sequences and the AMH sequences. We have examined these alleles and treated them as non-AMH AIMs, and we found that

phasing algorithms reasonably assigned these non-AMH AIMs to one haplotype as expected. These non-AMH AIMs are the most informative variants for our method and our method estimates these variants extremely well (S4.1.2).

Reviewer #1 (Remarks to the Author):

Comments to the authors:

After reading the replies of the authors I have further serious concerns about the validity of the method and the interpretation of the results from this study.

First, the authors focus on the model of Vernot et al (2014). There has been a substantial number of other studies since 2014 on archaic population substructure. In these studies, the estimated demographic parameters and admixture events differ at certain levels from the one of Vernot et al. The authors of this study do not provide any evidence that the model of Vernot (+archaic introgressions estimated by their method) explains better the observed data than any of the other methods (some of them provide support of such a statement by comparing the genetic diversity from simulated data). I am not convinced about the explanation why the amount of Denisovan ancestry is so small. In my opinion, this problem can be better explained by the demographic model that the authors are using. In other words, the benchmarking analysis that the authors conduct is, in my opinion, not considering all possible changes in effective population size, time of split and / or introgression requested for explaining current genetic variation.

Second, the argument that other studies use other types of statistics is also not convincing, particularly when the same results are obtained using many different statistics and demographic models. One possibility is that all these statistics are over estimating the estimates of admixture. However, each proposed statistic has been properly benchmarked towards complex simulations. This relates to the sentence: "It turned out that most of the site-based methods which were originally designed for analyzing genotyping data, such as D-Statistics, F-Statistics make a higher introgression estimation". In that case, I would suggest that the authors use these statistics to the same simulated data to show that they fail to identify the correct amount of admixture whereas the method that the authors propose do a proper assignment.

Third, the point of incomplete lineage sorting is also not properly explained. This point is particularly important because the authors are using the interpretation of a coalescence tree to train their methodology. However, two coalescence trees can show exactly the same shape even if in one case there has been archaic introgression and not in the other just by incomplete lineage sorting (i.e. some modern human lineages coalesce with archaic just by chance). Therefore, this approach of looking at the shape of the tree is not going to provide a ground truth for the introgressed sequences.

Fourth, the authors should have used msprime, which allows for archaic samples. Since the authors are using archaic samples, I assumed they had used msprime even if referring to ms. This is the reason why I asked about the code. In addition to considering ancient samples, msprime code is maintained and allows proper interpretation of the written model (please, see AJHG, Lessons Learned from Bugs in Models of Human History by Ragsdale et al). Moreover, I think that the approach that the authors apply to sample ancient samples, in addition to being outdated, can introduce artifacts in the final reported pseudoarchaic sequences just by the fact of shrinking in forward the demography from an ancient N_e to almost 0.

Fifth, if the method drops so dramatically after the suggested simulations (Comment 16: 2), I think this should be clearly stated in the manuscript. Furthermore, it undermines the value of the whole approach.

Reviewer #3 (Remarks to the Author):

Comments to the authors:

Review of: “Refining models of archaic-modern human admixture in Eurasia using ArchaicSeeker 2.0,” by Yuan et al.

1 July 2021

These authors describe a new version of their ArchaicSeeker software and use it to estimate the level and timing of archaic admixture into human populations throughout the world. Although I have several minor criticisms (see below), I think the method is sound and will recommend in favor of publication.

Previous reviewers expressed concern that the new estimates of Denisovan admixture into populations east of Wallace’s line are much lower than those of previous studies. However, Rogers and Bohlender [4] show that the estimators used in these previous studies exhibit strong biases when there is more than one source of archaic admixture. Their Fig. 4 shows that these estimates may be biased upwards by 600%. More recently, Rogers [3, Fig. 5] has confirmed these results in a simulation study. Since the populations of this region are thought to have received gene flow from two archaic sources, there is every reason to think that previous estimates of Denisovan admixture are too high.

Although I’m in favor of publication, I do have several concerns. First, I worry about the use of 1000-genomes data. The vcf files distributed by this project only include sites that are polymorphic in the project’s sample of modern humans. If other sites are excluded from the analysis, then we would miss sites at which the 1kgenomes sample was fixed for reference, but (say) Ust’-Ishim and Altai both carried a derived allele. Yet such sites are highly informative. This should induce a downward bias in estimates of archaic admixture. My guess is that this bias is small, but the authors ought to say something about it.

My second concern is detailed in items 10–11 below. Briefly, the likelihood function they define is incorrect. It reaches a maximum in the right place, so there should be no problem with maximum-likelihood estimates. The function is broader than it should be, however, and this will make likelihood-ratio tests (LRTs) less likely to reject false hypotheses. I’m not sure whether this problem affects the LRT that is used in this algorithm.

Finally, the model of linkage-disequilibrium decay is correct only in an infinite population, as explained in item 14 below. In small populations, this model exaggerates the rate at which LD decays and will cause a downward bias in estimates of the age of admixture events. I suspect this effect is small in populations of moderate size, but it would be wise to check that with computer simulations.

Comments on the main text

1. 518: “ArchaicSeeker 2.0 is the only available method that reconstructs multiple-wave intro-gression into present-day human populations.”

Several programs, including Legofit and Admixtools, will do this.

2. Fig 2: Why do deserts tend to end at centromeres? If they result from strong selection against deleterious archaic alleles, you would think that deserts would tend to be bounded by regions of high recombination. Yet recombination is low at centromeres, so the observed pattern is puzzling.

3. Fig 2: What is going on with the short arm of chromosome 21? There is a tan desert on that arm, but the remainder of the arm is white, indicating an absence of archaic admixture. Why isn't this entire arm considered a desert?

4. 930: It isn't conventional in mathematical notation to use "*" as a multiplication operator.
5. 991: The authors say that each cluster consisted of at least one SNP. But these clusters are based on r^2 , and you need two SNPs to calculate that. So it would seem that each cluster must contain at least two SNPs.

Comments on supplementary materials

6. S13: In estimating the rates of false and true positives, the authors should not have used the same simulated data sets that were used to train the HMM. It isn't clear from the text whether they did or not.
7. S13:¶3: Says the Vindija genome was used. This contradicts an earlier statement saying that Vindija was not used.
8. S17, Eqn 2.1: The definition implies that π_{ij} is the conditional probability, given allele frequencies in populations i and j , that two random gene copies, one drawn from i and the other from j , are copies of different alleles. The authors should say this in words.
9. S17, Eqn 2.2: The algebra would be simpler if you defined p_{ij} as the frequency of the derived allele. Then you would need neither η nor the inner summation in this equation. This is only a suggestion, which the authors should feel free to ignore.
10. S18: The authors say that "As we know, the genomic differences of any two populations (π_{ij}) follow a Poisson distribution of parameter $\lambda = L\mu D$, where D is the distance of these two populations in the matching model."

This would be true if the authors had actually chosen a random gene copy from population i and one from j at each SNP locus. But this is not what they did. The definition of π_{ij} (Eqn. 2.1) implies that it represents the average across all possible ways of choosing one gene copy from i and one from j . This average has the same mean as the Poisson distribution in Eqn. 2.6, but it has a smaller variance.

11. Equation 2.7 incorporates this Poisson distribution into an expression for the likelihood. It makes sense to maximize this likelihood, in spite of the incorrect distributional assumption, because its maximum is at the mean, and the mean is correct. However, they use a likelihood ratio test (LRT) later on in the algorithm. It was not clear to me whether the LRT was based on Eqn. 2.7. If it is, that test is incorrect; it would be less likely to reject false hypotheses than one based on the true likelihood. Either way, the authors should explain that (2.7) is not really the likelihood. It is a function whose maximum is at the same place as that of the likelihood.
12. S22, last line: Defines e as the end of a segment. But " e " is also used for exponentials in this manuscript. It would be best not to use that symbol for two unrelated purposes.
13. S23, just before section S2.4: The algorithm chooses the topology with the largest likelihood, in effect treating the topology as a parameter to be estimated. An alternative would be to average across topologies, weighting by likelihood. Why did you choose this approach? Perhaps this approach is necessary, because your goal is to classify each segment, and that involves estimating the topology. Right?

14. S23, equations 2.11 and 2.12: The model used here is only correct in an infinite population. In a very small population, very long segments of introgressed archaic chromosome would drift to fixation or loss so rapidly that recombination would not have time to break them up. Once fixed, these segments would remain long, because recombination would no longer occur. In general, the decay of linkage disequilibrium depends not only on the recombination rate, but also on population size [1, 2]. The authors should acknowledge that their model assumes a large population size and may exaggerate the rate at which LD decays in small populations. This may cause a downward bias in estimates of the time of introgression. To estimate that bias, it would be useful to do simulations of introgression into small populations.
15. As we know: At several places in the supplement, the authors use the phrase “as we know” to introduce theoretical ideas, which they believe to be well known, but which they do not cite. In several cases, these “well known” facts are not actually true. They should replace the unattributed assertions with citations to the original sources.

References

- [1] W. G. Hill and Alan Robertson. Linkage disequilibrium in finite populations. *Theoretical and Applied Genetics*, 38(6):226–231, 1968.
- [2] Alan R. Rogers. How population growth affects linkage disequilibrium. *Genetics*, 197(4):1329– 1341, August 2014.
- [3] Alan R. Rogers. Legofit: Estimating population history from genetic data. *BMC Bioinformatics*, 20:526, 2019.
- [4] Alan R. Rogers and Ryan J. Bohlender. Bias in estimators of archaic admixture. *Theoretical Population Biology*, 100:63–78, March 2015. ISSN 0040-5809.

Authors' Response to Reviewer #1 (Remarks to the Author):

After reading the replies of the authors I have further serious concerns about the validity of the method and the interpretation of the results from this study.

Authors' Response:

We thank the reviewer for the additional comments on our manuscript. We have made substantially extra analyses and updated the manuscript accordingly; we hope we have fully addressed all the concerns raised by the reviewer.

Comment 1: First, the authors focus on the model of Vernot et al (2014). There has been a substantial number of other studies since 2014 on archaic population substructure. In these studies, the estimated demographic parameters and admixture events differ at certain levels from the one of Vernot et al. The authors of this study do not provide any evidence that the model of Vernot (+archaic introgressions estimated by their method) explains better the observed data than any of the other methods (some of them provide support of such a statement by comparing the genetic diversity from simulated data). I am not convinced about the explanation why the amount of Denisovan ancestry is so small. In my opinion, this problem can be better explained by the demographic model that the authors are using. In other words, the benchmarking analysis that the authors conduct is, in my opinion, **not considering all possible changes in effective population size**, time of split and / or introgression requested for explaining current genetic variation.

Authors' Response:

We thank the reviewer for this comment. The simulation studies of our paper are based on the model of Vernot et al (2014) as well as that of a more recent study (Skov et al Nature, 582:78-73, 2020). Furthermore, we would like to clarify that we have considered a number of different parameter values and their combinations in our simulation studies, including split time (276 ~ 403 kya for Denisovan and 75 ~ 115 kya for Neanderthal, **Figure S4.2**), introgression time (18 ~ 82 kya, **Figure S4.3**) and introgression proportion (0.4% ~ 4%, **Figure S4.4**). Although it is not realistic to simulate ALL POSSIBLE changes of these parameters, we believe our studies have covered the key parameters which could have substantial influences on our proposed methods in a reasonable range, and the simulation outcomes indicate that our method is less influenced by these changes, and hence the results are acceptable with different demographic parameters.

In addition to the model by Vernot et al., we also performed a series of additional simulations based on the model specified in a more recent study (Skov et al Nature, 582:78-73, 2020) in our first-round replies, which may have yet caught the attention of the reviewer. In Skov's model, the effective population sizes are much different from those in Vernot's model. However, our simulation results (see **Figure R1 in the first response**) indicated still good performance of our method: The mean value of Precision is 94.72% (92.47% ~ 96.06%), TPR is 84.37% (80.45% ~ 88.01%), and FPR is 0.50% (0.32% ~ 0.76%).

Regarding the particular concern of the reviewer, i.e., "why the amount of Denisovan ancestry is so small", we believe the reason is that the overestimation based on D statistics in previous studies, rather than the underestimation of our method. It turned out that the D statistics overestimated the archaic introgression for 5-10 folds based on our simulation analysis (see also details below). As Reviewer #3 also pointed out, "Rogers and Bohlender [4] showed that the estimators used in these previous studies exhibit strong biases when there is more than one source of archaic admixture. Their Fig. 4 shows that these estimates may be biased upwards by 600%. More recently, Rogers [3, Fig. 5] has confirmed these results in a simulation study."

Comment 2: Second, the argument that other studies use other types of statistics is also not convincing, particularly when the same results are obtained using many different statistics and demographic models. One possibility is that all these statistics are over estimating the estimates of admixture. However, **each proposed statistic has been properly benchmarked towards complex simulations**. This relates to the sentence: "It turned out that most of the site-based methods which were originally designed for analyzing genotyping data, such as **D-Statistics, F-Statistics** make a higher introgression estimation". In that case, I would suggest that the authors use these statistics to the same simulated data to show that they fail to identify the correct amount of admixture whereas the method that the authors propose do a proper assignment.

Authors' Response:

We thank the reviewer for this comment. We agree that D Statistics and F Statistics are widely used in relevant fields. However, we think it might be premature to suggest that these two statistics are "properly benchmarked towards complex simulations" for estimating the archaic introgression proportion. To our best knowledge, we have yet found any benchmark studies in the literature which explicitly used D or F to estimate the archaic introgression proportion. As a matter of fact, it turned out that the D statistics overestimated the archaic introgression for 5-10 folds based on our simulation analysis (see also details below).

As a step towards assessing the effectiveness of these statistics for estimating the archaic introgression proportion, we used the D Statistics to calculate the introgression proportion of Neanderthal and Denisovan based on simulated data. We used msprime to simulate five populations approximating Vindija Neanderthal, Denisovan, Chimpanzee, Africans, and non-Africans mainly following the Skov's model (Skov et al Nature, 582:78-73, 2020). The sample size of each population is 1, 1, 1, 1, 100, and 100, respectively. In addition to the parameters in Skov's model, we here also included detailed demographic parameters simulating population Chimpanzee: the population size of the common ancestor of Chimpanzee and human was 52,000 and the population size of Chimpanzee was 15,000 (Feng-Chi Chen & Wen-Hsiung Li, American Journal of Genetics, DOI: 10.1086/318206). The split time of Chimpanzee and human was set as 6,000,000 years (Yong-Jin Won & Jody Hey, Molecular Biology and Evolution, <https://doi.org/10.1093/molbev/msi017>). The simulated chromosome length was 30Mb and we assumed the recombination rate as 1.2 cM/Mb and mutation rate as 1.2e-8 per generation per site, along with a 29-year generation time. The introgression proportions were set as 0.5% and 1% 45,000 years ago. AdmixTools v 7.0.2 was used to calculate the D statistics and the equation was

$$\frac{D(\text{African, Test, Introgressor, Chimpanzee})}{D(\text{African, Introgressor}_1, \text{Introgressor}_2, \text{Chimpanzee})}$$

where *Introgressor*₁ and *Introgressor*₂ were randomly chosen subsets of half of the Introgressor (Neanderthal or Denisovan) (Nathan K. Schaefer et al. Science Advances, 10.1126/sciadv.abc0776). It turned out that the D statistics overestimated for 5-10 folds based on the estimation results (**Figure R1 and Table R1, R2**).

We also estimated the introgression proportion with ArchaicSeeker2. The Denisovan introgression proportion is underestimated. The reason was that about half of the Denisovan introgression sequences were matched the corrected ancestry and ~ 40% were matched to other ancestries, most of which are the ancestor of Denisovan and Neanderthal. The Neanderthal introgression proportion is slightly underestimated for the "0.5% introgression scenario" and is slightly overestimated for the "1% introgression scenario". As we discussed in the supplementary (**S4.1, Figure S4.26-4.32**), ancestry mismatch happened more frequently in Denisovan introgressed sequences and would cause the underestimation of the Denisovan introgression proportion. Compared with the extreme overestimation of D statistics, the estimation based on ArchaicSeeker2 is much reliable and accurate.

Figure R1. Archaic introgression estimated by ArchaicSeeker2 and D statistics. Introgression proportions were estimated through the simulation data. The black dash line indicated the ground truth introgression proportion.

Table R1. The mean of estimated introgression proportion under the Denisovan introgression scenario

Introgression Proportion	ArchaicSeeker2		D statistics
	Denisovan	Others	
0.5%	0.245%	0.171%	2.55%
1%	0.516%	0.385%	3.7%

Table R2. The mean of the estimated introgression proportion under the Neanderthal introgression scenario

Introgression Proportion	ArchaicSeeker2		D statistics
	Neanderthal	Others	
0.5%	0.384%	0.044%	3.27%
1%	1.107%	0.066%	11.4%

Finally, we would like to point out that we are not alone on having reservation regarding the effectiveness of D Statistics for estimating the archaic introgression proportion. For example, the authors in a recent study (Nathan K. Schaefer et al. Science Advances, 10.1126/sciadv.abc0776) compared the introgression proportion estimated by D Statistics and their method SARGE, and they didn't observe that high proportion of Denisovan ancestry in Oceania with their method while D Statistics suggested a higher introgression proportion (Figure R2).

Figure R2. Archaic introgression proportion in the worldwide populations (Figure 3, Nathan K. Schaefer et al. Science Advances, 10.1126/sciadv.abc0776)

Comment 3: Third, the point of incomplete lineage sorting is also not properly explained. This point is particularly important because the authors are using the interpretation of a coalescence tree to train their methodology. However, **two coalescence trees can show exactly the same shape** even if in one case there has been archaic introgression and not in the other just by incomplete lineage sorting (i.e. some modern human lineages coalesce with archaic just by chance). Therefore, this approach of looking at the shape of the tree is not going to provide a ground truth for the introgressed sequences.

Authors' Response:

We thank the reviewer for this comment. We agree that incomplete lineage sorting (ILS) is a challenging issue for studying archaic introgression. To address this problem at the methodology level, the proposed archaic introgression inference method, ArchaicSeeker2, not only consider the tree topology, but also the branch length of the phylogenetic tree (matching algorithm) and the introgressed sequences length (removing the extreme shorter sequences) to mitigate the influence ILS.

Indeed, in our previous simulations with ms, we cannot distinguish the ILS sequences from the really introgressed sequences. Following the suggestion of the reviewer, we used msprime to generate simulation data and used their built-in module to trace the ground truth introgressed sequences. We used the simulation model from Skov et al (Nature, 582:78-73, 2020) study. Both ms and msprime were used to generate the simulation data under Skov's model. We generated simulation data with msprime and ms, respectively and repeated them 100 times (see the response to Comment 4 for the simulation details).

The results on the datasets simulated by ms and msprime are almost identical. For ms simulations, the mean value of Precision is 94.72% (92.47% ~ 96.06%), TPR is 84.37% (80.45% ~ 88.01%) and FPR is 0.50% (0.32% ~ 0.76%). For msprime simulations, the mean value of Precision is 94.52% (92.59% ~ 95.81%), TPR is 86.05% (82.13% ~ 89.19%) and FPR is 0.52% (0.30% ~ 0.86%) (Figure R3). The TPR of msprime simulations (86.05%) are slightly higher than that of the ms simulations (84.37%), which including

ILS sequences. That indicated our method could remove certain ILS sequences, as part of the design goals of our method.

Figure R3. Length based evaluation with ms and msprime under Skov's simulation scenario. Comparison between the inferred introgressed segments and the ground truth segments under the scenarios described in Skov et al 2020. The y-axis represents the summary statistics of precision, TPR and FPR, respectively. The x-axis represents the two simulation methods (ms and msprime) that are used to generate the data.

In our method, we used the Hidden Markov Model to model the distribution of the archaic introgression sequences on the modern non-African human genomes. Unlike site-based methods, such as D statistics and F statistics, we considered not only the nucleotide / SNP information, but also the length of each introgressed sequence. The ILS should affect the site-based method much greater than our method, which might also be part of the reasons that comparatively higher over estimations observed on site-based methods.

Comment 4: Fourth, the authors should have used msprime, which allows for archaic samples. Since the authors are using archaic samples, I assumed they had used msprime even if referring to ms. This is the reason why I asked about the code. In addition to considering ancient samples, msprime code is maintained and allows proper interpretation of the written model (please, see AJHG, Lessons Learned from Bugs in Models of Human History by Ragsdale et al). Moreover, I think that the approach that the authors apply to sample ancient samples, in addition to being outdated, can introduce artifacts in the final reported pseudoarchaic sequences just by the fact of shrinking in forward the demography from an ancient Ne to almost 0.

Authors' Response:

We thank the reviewer for this comment. Following the suggestion of the reviewer, we have managed to overcome some technical challenges to incorporate msprime in our study (see the new section S4.3 in the Supplementary Information), including generating simulation data and using their built-in module to trace the ground truth introgressed sequences. Based on the simulation model from Skov et al (Nature, 582:78-73, 2020) study, we used both ms and msprime to generate simulation datasets under Skov's model, with msprime code obtained directly from their study (<https://github.com/LauritsSkov/ArchaicSimulations>). We simulated 10 Mb sequences of 100 Africans (200 haplotypes), 100 Non-Africans (200 haplotypes), 1 Neanderthal (2 haplotypes) and 1 Denisovan (2 haplotypes) with msprime and ms, respectively, and repeated them 100 times.

The results of the two software are almost identical. For ms simulations, the mean value of Precision is 94.72% (92.47% ~ 96.06%), TPR is 84.37% (80.45% ~ 88.01%) and FPR is 0.50% (0.32% ~ 0.76%). For msprime simulations, the mean value of Precision is 94.52% (92.59% ~ 95.81%), TPR is 86.05% (82.13% ~ 89.19%) and FPR is 0.52% (0.30% ~ 0.86%). We note that the TPR of msprime simulations (86.05%) are slightly higher than that of the ms simulations (84.37%), as ILS sequences are excluded in msprime simulations. This indicated our proposed method could remove certain ILS sequences, as our method designed.

Although, there might be some reservation on ms, especially when people applied this method to complex history model, we did not observe significant differences between the results of the two simulation methods in our study. Therefore, we don't think the "outdated" approach introduce "artifacts in the final reported pseudoarchaic sequences" and we believe our previous simulation framework is also robust and reliable.

Comment 5: Fifth, if the method drops so dramatically after the suggested simulations (Comment 16: 2), I think this should be clearly stated in the manuscript. Furthermore, it undermines the value of the whole approach.

Authors' Response:

We thank the reviewer for this comment. We would like to emphasize that the suggested simulations mentioned in this comment are proposed in Comment 16 of the reviewer, that is, to distribute at random the fragments in the simulated data by splitting them into 10kb bins and random shuffling. In the response of Comment 16 of the first-round revision, we conducted suggested simulations and observed a clear drop of TPR and precision and an increase of FPR in the random shuffling data, i.e., TPR is only 48.5% (35.4 ~ 51.5%), precision is 81.1% (78.4 ~ 88.7%), and FPR is 0.236% (0.093 ~ 0.285%), compared with previous TPR 90.8%, precision 93.7%, and FPR 0.124%.

The significant drops in the results are not a surprise for us as our methods are designed for working with true human genomic data in which randomness is resulted from various evolutionary forces working together and hence is intrinsically different from the 'uniform' randomness in the suggested simulations in Comment 16. More precisely, unlike site-based and several other archaic introgression inference methods, ArchaicSeeker2 utilize information on both nucleotide differences and introgressed sequences length. One key model assumption in our method is that the lengths of introgressed sequences follow an exponential distribution, which it is true when the effective population size is not extremely small. In the simulations according to Comment 16, the genomic data is shuffled randomly, which changed the length distribution of these introgressed sequences in an abrupt and unrealistic way.

To confirm this, we subsequently did a series of simulation with different bin sizes: 10k, 20k, 50k, 100k, 200k and 500k. As expected, longer bin size has much less unfavorable effects on the results. The average introgressed sequences are ~50kb. If we used a longer bin size, more introgressed sequences were likely not be splitted by the shuffle and the performance is much better (Figure R5).

To sum up, the accuracy of our method relies on the accuracy of both SNP level information and introgressed length information. The drops of the results on the suggested simulations, albeit alarming at the first sight, actually demonstrate that the length information is utilized successfully in our method, as planned.

Figure R5. Length based evaluation of genomic segment shuffle with different bin size. The y-axis represents the summary statistics of precision, TPR and FPR, respectively. The x-axis represents the bin size used to shuffle the data. The dash red line indicated the precision, TPR and FPR before shuffle.

Authors' Response to Reviewer #3 (Remarks to the Author):

These authors describe a new version of their ArchaicSeeker software and use it to estimate the level and timing of archaic admixture into human populations throughout the world. Although I have several minor criticisms (see below), I think the method is sound and will recommend in favor of publication.

Previous reviewers expressed concern that the new estimates of Denisovan admixture into populations east of Wallace's line are much lower than those of previous studies. However, Rogers and Bohlender [4] show that the estimators used in these previous studies exhibit strong biases when there is more than one source of archaic admixture. Their Fig. 4 shows that these estimates may be biased upwards by 600%. More recently, Rogers [3, Fig. 5] has confirmed these results in a simulation study. Since the populations of this region are thought to have received gene flow from two archaic sources, there is every reason to think that previous estimates of Denisovan admixture are too high.

Authors' Response:

We thank the reviewer for the positive comments on our manuscript and pointed out additional studies in the literature concerning the estimation of Denisovan admixture. We have incorporated them into the updated version and below are our responses to the specific concerns.

Comment 1: Although I'm in favor of publication, I do have several concerns. First, I worry about the use of 1000-genomes data. The vcf files distributed by this project only include sites that are polymorphic in the project's sample of modern humans. If other sites are excluded from the analysis, then we would miss sites at which the 1kgenomes sample was fixed for reference, but (say) Ust'-Ishim and Altai both carried a derived allele. Yet such sites are highly informative. This should induce a downward bias in estimates of archaic admixture. My guess is that this bias is small, but the authors ought to say something about it.

Authors' Response:

We thank the reviewer for this comment. We agreed these variants are of highly informative. We counted the number of sites that are carrying alternative alleles in both Altai Neanderthal and Ust'-Ishim. There are 957,244 such sites in total and we can find 930,806 (97.2%) of them from the 1000 Genomes data. Therefore, there might be some bias, but which, as you mentioned, must be small.

Comment 2: My second concern is detailed in items 10-11 below. Briley, the likelihood functions they define is incorrect. It reaches a maximum in the right place, so there should be no problem with maximum-likelihood estimates. The function is broader than it should be, however, and this will make likelihood-ratio tests (LRTs) less likely to reject false hypotheses. I'm not sure whether this problem affects the LRT that is used in this algorithm.

Authors' Response:

We thank the reviewer for these comment. We agree with the reviewer's opinion on our likelihood function and maximum-likelihood estimates. We use the Poisson distribution in Eqn. 2.6 to approximate the true distribution, which maybe be of certain simplification, but there should be no problem with maximum-likelihood estimates. In addition, the LRT in our method was not based on Eqn. 2.7, which was used to select a proper admixture model (i.e., the number of admixture waves) to describe the history of archaic introgression. The LRT in our method was based on the length distribution of introgressed segments, which is a mixed exponential distribution.

Comment 3: Finally, the model of linkage-disequilibrium decay is correct only in an infinite population, as explained in item 14 below. In small populations, this model exaggerates the rate at which LD decays and will cause a downward bias in estimates of the age of admixture events. I suspect this effect is small in populations of moderate size, but it would be wise to check that with computer simulations.

Authors' Response:

We fully understand the reviewer’s concerns. We appreciate that the patterns of LD decay can deviate from the described model when the population size is very small. To evaluate the potential effect, we used the forward-time simulator *AdmixSim2* (in revision in BMC Bioinformatics, <https://github.com/Shuhua-Group/AdmixSim2>) to simulate a hybrid isolation (HI) model under different population sizes (100, 200, 500, 1000, 2000, 3000, 4000, 5000, 10000). The simulated generation was 2,000 and the admixture proportion of two ancestors were 2% and 98%, respectively. We then compared the distribution of simulated segments (red points) from the minor contributor and theoretical distribution (black line). From the results, the simulated segments under the conditions of population size less than 1,000 deviated from the theoretical distribution and the mean length tended to be much larger. However, when the population size was larger than or equal to 1,000, the simulated distributions approximated very closely to the theoretical one. Therefore, the described model can be inappropriate when the population size is less than 1,000, but the population size used in most population genetic studies is approximately 5,000 (the effective population size of non-African test population). Thus, we believe the effect of population size is minor in most application scenarios.

Figure R6. The introgressed sequences length distribution with different effective population size. The black solid line is the theoretical length distribution of introgressed sequences and the red dots stand for the length distribution of the simulated sequences.

Comments on the main text

Comment M1: 518: “ArchaicSeeker 2.0 is the only available method that reconstructs multiple-wave introgression into present-day human populations.”

Several programs, including Legofit and Admixtools, will do this.

Authors' Response:

We thank the reviewer for this comment. We have updated this sentence and added the two methods in our introduction and discussion sections.

Comment M2: Fig 2: Why do deserts tend to end at centromeres? If they result from strong selection against deleterious archaic alleles, you would think that deserts would tend to be bounded by regions of high recombination. Yet recombination is low at centromeres, so the observed pattern is puzzling.

Authors' Response:

We thank the reviewer for this comment. Indeed, we are also puzzled by this observation. One plausible reason to explain the deserts of archaic sequence in modern non-African genomes is a result of some strong negative selection. If we assumed a simple additive model, those sequences enriched with deleterious archaic sties should be likely removed by the negative selection. However, so far, we do not have sufficient information or evidence to support this explanation.

Comment M3: Fig 2: What is going on with the short arm of chromosome 21? There is a tan desert on that arm, but the remainder of the arm is white, indicating an absence of archaic admixture. Why isn't this entire arm considered a desert?

Authors' Response:

We thank the reviewer for this comment. Actually, chromosome 21 is a very tricky chromosome. We used the genomic gap annotation from UCSC. The related gaps are as follow,

#bin	chrom	chromStart	chromEnd	ix	n	size	type	bridge
1	chr21	5211193	9411193	3	N	4200000	contig	no
9	chr21	10000	5211193	2	N	5201193	short_arm	no
10	chr21	11288129	14288129	22	N	3000000	centromere	no

There is a large region (chr21: 5211193-9411193) of type “contig” and there is no genetic data in this region. We didn't define this “contig” region as desert because there are no sufficient data to be analyzed.

Comment M4: 930: It isn't conventional in mathematical notation to use “*” as a multiplication operator.

Authors' Response:

Following the suggestion, we have replaced “*” with “×” in our manuscript.

Comment M5: 991: The authors say that each cluster consisted of at least one SNP. But these clusters are based on r^2 , and you need two SNPs to calculate that. So it would seem that each cluster must contain at least two SNPs.

Authors' Response:

There seems to be a misunderstanding about the methodology, which we hope to clarify through our responses below and by revising the text in the manuscript. The r^2 was calculated for SNPs within a 500-kb sliding window, which consisted of at least two SNPs. SNPs were assigned to one cluster if they are in high LD ($r^2 > 0.8$). However, if one could not be tagged by any other SNPs, it would represent an independent cluster. Therefore, we could find some of the clusters contains only one SNP. Consequently, the sentence in line 989-993 has been revised accordingly as follows:

“To test the enrichment of the eQTLs in the archaic alleles, we first estimated LD for SNPs within a 500-kb sliding window using PLINK v1.9, and then clustered the genome-wide SNPs with high LD ($r^2 > 0.8$) in each population group. Each of the SNPs that could not be tagged by any others represented a single cluster. We further defined one cluster as an archaic locus ...”

Comments on supplementary materials

Comment S6: S13: In estimating the rates of false and true positives, the authors should not have used the same simulated data sets that were used to train the HMM. It isn't clear from the text whether they did or not.

Authors' Response:

We thank the reviewer for this comment. In our method, we use a modified expectation–maximization (EM) algorithm to estimate the HMM parameters and the Viterbi Algorithm to infer the hidden states of each SNP and then find the candidate archaic introgressed segments. These steps are required for each simulated dataset. Based on the lengths of the inferred introgressed segments and ground-truth introgressed segments, we calculated the rates of false and true positives for each simulated dataset. Among 14,400 simulations, the median value of the precision was 93.0% (95% CI, 89.4%–95.9%), that of the TPR was 90.4% (95% CI, 84.1%–94.1%), and that of the FPR was 0.14% (95% CI, 0.07%–0.22%). We have modified the corresponding part of supplementary.

Comment S7: S13:¶3: Says the Vindija genome was used. This contradicts an earlier statement saying that Vindija was not used.

Authors' Response:

We thank the reviewer for pointing out this typo. We have corrected it.

Comment S8: S17, Eqn 2.1: The definition implies that π_{ij} is the conditional probability, given allele frequencies in populations i and j , that two random gene copies, one drawn from i and the other from j , are copies of different alleles. The authors should say this in words.

Authors' Response:

We thank the reviewer for this suggestion. We have modified the definition of π_{ij} in the supplementary.

Comment S9: S17, Eqn 2.2: The algebra would be simpler if you defined p_{ij} as the frequency of the derived allele. Then you would need neither η_k^l nor the inner summation in this equation. This is only a suggestion, which the authors should feel free to ignore.

Authors' Response:

We thank the reviewer for this suggestion. We have modified the definition of Ω_i in the supplementary. With this suggestion, the equation is indeed much simpler.

Comment S10: S18: The authors say that “As we know, the genomic differences of any two populations (π_{ij}) follow a Poisson distribution of parameter $\lambda = L\mu D$, where D is the distance of these two populations in the matching model.”

This would be true if the authors had actually chosen a random gene copy from population i and one from j at each SNP locus. But this is not what they did. The definition of π_{ij} (Eqn. 2.1) implies that it represents the average across all possible ways of choosing one gene copy from i and one from j . This average has the same mean as the Poisson distribution in Eqn. 2.6, but it has a smaller variance.

Authors' Response:

We thank the reviewer for this comment. We agree with the reviewer's opinion. π_{ij} has the same mean as the Poisson distribution in Eqn. 2.6, but it has a smaller variance. The variance of π_{ij} is related to the sample size. For simplicity, we use the Poisson distribution in Eqn. 2.6 to approximate the true distribution. As the reviewer said in comment 11, it makes sense and has little influence on the maximum-likelihood estimates.

Comment S11: Equation 2.7 incorporates this Poisson distribution into an expression for the likelihood. It makes sense to maximize this likelihood, in spite of the incorrect distributional assumption, because its maximum is at the mean, and the mean is correct. However, they use a likelihood ratio test (LRT) later on in the algorithm. It was not clear to me whether the LRT was based on Eqn. 2.7. If it is, that test is incorrect; it would be less likely to reject false hypotheses than one based on the true likelihood. Either way, the authors should explain that (2.7) is not really the likelihood. It is a function whose maximum is at the same place as that of the likelihood.

Authors' Response:

We thank the reviewer for this comment. We agree with the reviewer's opinion on our likelihood function and maximum-likelihood estimates. We use the Poisson distribution in Eqn. 2.6 to approximate the true distribution, which maybe lead to certain simplification, but there should be no problem with maximum-likelihood estimates. In addition, the LRT in our method was not based on Equation 2.7, which was used to select a proper admixture model (the number of admixture waves) to describe the history of archaic introgression. The LRT was based on the length distribution of introgressed segments, which is a mixed exponential distribution.

Comment S12: S22, last line: Defines e as the end of a segment. But "e" is also used for exponentials in this manuscript. It would be best not to use that symbol for two unrelated purposes.

Authors' Response:

We thank the reviewer for this suggestion. In the revised version we used "z" to describe the end of one segment, instead of "e".

Comment S13: S23, just before section S2.4: The algorithm chooses the topology with the largest likelihood, in effect treating the topology as a parameter to be estimated. An alternative would be to average across topologies, weighting by likelihood. Why did you choose this approach? Perhaps this approach is necessary, because your goal is to classify each segment, and that involves estimating the topology. Right?

Authors' Response:

We thank the reviewer for this comment. The reviewer is correct in saying that selecting the topology with the largest likelihood is used in the matching algorithm. At this stage, our goal is to classify each sequence to a proper ancestry. For history inference, we used another model based on the length distribution of those introgressed sequences.

Comment S14: S23, equations 2.11 and 2.12: The model used here is only correct in an infinite population. In a very small population, very long segments of introgressed archaic chromosome would drift to fixation or loss so rapidly that recombination would not have time to break them up. Once fixed, these segments would remain long, because recombination would no longer occur. In general, the decay of linkage disequilibrium depends not only on the recombination rate, but also on population size [1, 2]. The authors should acknowledge that their model assumes a large population size and may exaggerate the rate at which LD decays in small populations. This may cause a downward bias in estimates of the time of introgression. To estimate that bias, it would be useful to do simulations of introgression into small populations.

Authors' Response:

We thank the reviewer for this comment. We have detailed our response to this comment in our reply to Comment 3 in the previous section.

Comment 15: As we know: At several places in the supplement, the authors use the phrase “as we know” to introduce theoretical ideas, which they believe to be well known, but which they do not cite. In several cases, these “well known” facts are not actually true. They should replace the unattributed assertions with citations to the original sources.

Authors' Response:

We thank the reviewer for this comment. As suggested, we have added proper citations to these places.

Reviewers' Comments:

Reviewer #1:

Remarks to the Author:

The authors have addressed all my previous concerns and comments. Given their results, I do not have further comments.

Reviewer #3:

Remarks to the Author:

This is an excellent paper. The authors have responded to all concerns expressed by me and by the other reviewer. I recommend in favor of publication.

Point-to-point response to the reviewers' comments

Reviewer #1 (Remarks to the Author):

Comment 1: The authors have addressed all my previous concerns and comments. Given their results, I do not have further comments.

Response: We thank the reviewer for the positive comments and support!

Reviewer #3 (Remarks to the Author):

Comment 2: This is an excellent paper. The authors have responded to all concerns expressed by me and by the other reviewer. I recommend in favor of publication.

Response: We thank the reviewer for the positive comments and support!